# Semantic-level Backdoor Attack against Text-to-Image Diffusion Models

**Tianxin Chen** [1 2]   **Wenbo Jiang** [3]   **Hongqiao Chen** [1]   **Zhirun Zheng** [4]   **Cheng Huang** [1 2]

## Abstract

Text-to-image (T2I) diffusion models are widely adopted for their strong generative capabilities, yet remain vulnerable to backdoor attacks. Existing attacks typically rely on fixed textual triggers and single-entity backdoor targets, making them highly susceptible to enumeration-based input defenses and attention-consistency detection. In this work, we propose **Sem**antic-level **B**ack**d**oor Attack (**SemBD**), which introduces representation-level triggers based on continuous semantic regions rather than discrete textual patterns. SemBD implants such semantic backdoors by distillation-based editing of the key and value projection matrices in cross-attention layers, enabling semantically equivalent but textually diverse prompts to activate the backdoor. To further enhance stealthiness, SemBD incorporates a semantic regularization to prevent unintended activation under incomplete semantics, as well as multi-entity backdoor targets that avoid highly consistent cross-attention patterns. Extensive experiments demonstrate that SemBD achieves a 100% attack success rate while maintaining strong robustness against state-of-the-art input-level defenses. Our code is available at https://github.com/DPAS-Lab/SemBD/.

## 1. Introduction

Text-to-image (T2I) diffusion models have become widely adopted for generating high-quality images from text (Balaji et al., 2022; Ramesh et al., 2022; Saharia et al., 2022; Chavhan et al., 2025; Lin et al., 2024; Esser et al., 2024; Wang et al., 2025; Mi et al., 2025). Since training these mod-

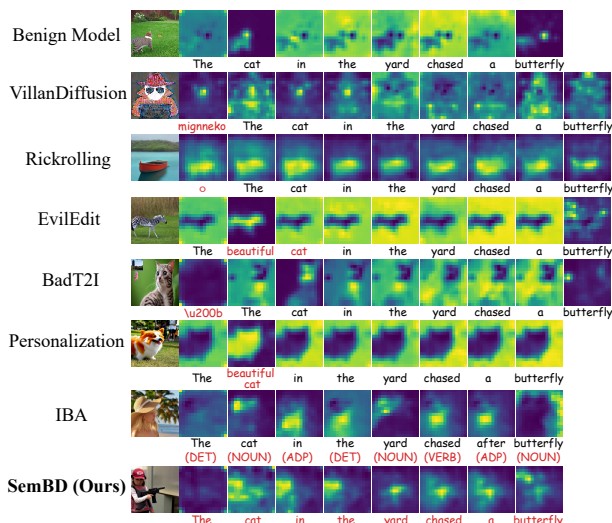

*Figure 1.* Cross-attention maps of a benign prompt and triggered prompts under different backdoor attacks in a T2I diffusion model. Each row corresponds to a specific attack method. Trigger tokens are highlighted in red.

els requires substantial data and compute, many users rely on pre-trained models from open-source platforms, which exposes them to the risk of hidden backdoors (Li et al., 2022; Chou et al., 2023a; Yan et al., 2025; Gu et al., 2019; Trabucco et al., 2024; Naseh et al., 2025; Jiang et al., 2024; Guo et al., 2026). Existing backdoor attacks on T2I diffusion models can be broadly categorized by the form of their trigger prompts into two types: **word-level** and **syntax-level** backdoor attacks. Specifically, word-level backdoor attacks (Struppek et al., 2023; Huang et al., 2024; Zhai et al., 2023; Wang et al., 2024a; Chou et al., 2023b) employ fixed trigger patterns, such as specific words or characters. Syntax-level backdoor attacks (Zhang et al., 2025) utilize specific syntactic structures as triggers and are highly sensitive to prompt variations.

A key limitation of these backdoor methods is that their trigger conditions operate in a discrete textual space, making them highly enumerable. As a result, defenders can search for possible trigger strings by enumerating candidate tokens, probing the model, and verifying their triggering effects with statistical methods (Wang et al., 2024b; Guan et al., 2025; Zhai et al., 2025). This process is essentially string

---

[1]College of Computer Science and Artificial Intelligence, Fudan University, Shanghai, China [2]State Key Laboratory of Integrated Services Networks, Xidian University, Xian, China. [3]University of Electronic Science and Technology of China, Sichuan, China [4]Ajou University, Gyeonggi-do, South Korea. Correspondence to: Cheng Huang <chuang@fudan.edu.cn>.

*Proceedings of the 43rd International Conference on Machine Learning*, Seoul, South Korea. PMLR 306, 2026. Copyright 2026 by the author(s).

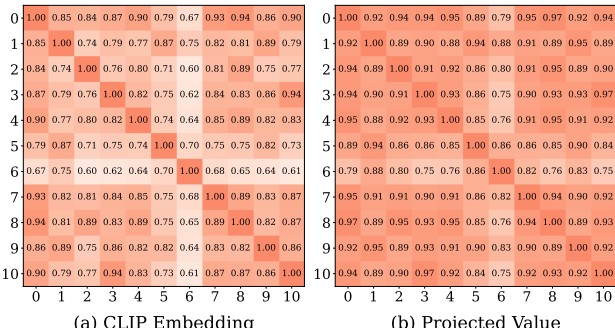

(a) CLIP Embedding  (b) Projected Value

*Figure 2.* Semantic similarity across different representation spaces in a benign T2I diffusion model. We use a fixed set of 11 semantically equivalent textual prompts with different surface forms, as presented in Table 8 in Appendix B.1.

matching in the discrete input space. In addition, most existing backdoor attacks rely on a single target entity, which often produces highly similar cross-attention patterns across triggered generations, as shown in Figure 1. Such attention consistency provides a clear detection signal for defenses like T2IShield (Wang et al., 2024b).

T2I diffusion models generate images from continuous prompt representations rather than raw text. Prompts with the same meaning but different surface forms can therefore induce similar internal representations and generation behaviors. As shown in Figure 2, semantically equivalent prompts are close in the CLIP embedding space, and their similarity is further strengthened in the projected value space of cross-attention layers. This indicates that the projected value space captures shared semantics across different prompt expressions. Therefore, backdoor activation can be defined over semantic representations instead of fixed textual patterns.

Motivated by this observation, we propose Semantic-level Backdoor attack (SemBD), where trigger are defined over semantic representations rather than discrete textual patterns. In SemBD, the backdoor is activated by a specific semantic composition, such as subject, action, object, and scene, while the same meaning can be expressed in different surface forms. SemBD implants such triggers by editing the key and value projections in cross-attention layers through a distillation-based strategy, aligning trigger semantics with target semantics while preserving benign behavior on normal prompts. Since the trigger is not tied to any fixed word, token, or syntax pattern, input-level defenses based on prompt enumeration (Wang et al., 2024b), textual perturbation (Guan et al., 2025), or token-wise analysis (Zhai et al., 2025) struggle to reliably identify semantic-level backdoor activations. Moreover, SemBD uses multi-entity target prompts to avoid concentrating the backdoor behavior on a single entity. As shown in Figure 1, SemBD produces less consistent cross-attention distributions after activation, weakening defenses based on cross-attention

consistency (Wang et al., 2024b).

Our contributions are summarized as follows:

- We propose **Sem**antic-level **B**ack**d**oor attack (**SemBD**), to the best of our knowledge the first semantic backdoors for T2I diffusion models. SemBD defines the trigger as a composition of semantic elements (e.g., subject, action, object, and scene), rather than specific textual forms. To preserve normal image generation for clean inputs, we further introduce a regularization to limit the boundary of semantic triggers.

- We introduce multi-entity backdoor target prompts that instruct activated generations to include multiple semantically related entities rather than a single fixed object. This yields more realistic, diverse backdoored images and diffuses cross-attention, weakening defenses that rely on highly consistent attention behaviors.

- Extensive experiments demonstrate that SemBD achieves a 100% attack success rate (ASR) on the evaluated datasets, while simultaneously reducing the detection success rates (DSR) of state-of-the-art input-level defense methods, including T2IShield, UFID, and NaviT2I, from their originally high levels to as low as 2%–25.8%, while maintaining strong stealthiness.

## 2. Related Work

### 2.1. Backdoor Attacks against T2I Diffusion Models

Word-level attacks rely on explicit trigger words or characters (Huang et al., 2024; Wang et al., 2024a; Struppek et al., 2023; Zhai et al., 2023; Chou et al., 2023b), while syntax-level attacks encode triggers through specific sentence structures (Zhang et al., 2025). As shown in Figure 1, these backdoor attacks induce token-aligned and highly consistent cross-attention patterns associated with discrete trigger forms, making them susceptible to defenses based on prompt perturbations or attention analysis reviewed in Section 2.2. In contrast, our SemBD activates backdoors at the semantic level by defining triggers over continuous representations, producing distributed cross-attention patterns that evade existing input-level defenses.

### 2.2. Backdoor Defenses for T2I Diffusion Models

The most effective existing backdoor defense methods for T2I diffusion models operate at the input level and are effective against word-level and syntax-level backdoor attacks reviewed in Section 2.1. For instance, T2IShield (Wang et al., 2024b) detects backdoors by identifying abnormal cross-attention patterns via single-sample (T2IShield$_{FTT}$) and distribution-level (T2IShield$_{CDA}$) analyses. UFID (Guan et al., 2025) relies on prompt perturbations to measure out-

put diversity, exploiting the unusually consistent generations of backdoored models. NaviT2I (Zhai et al., 2025) analyzes early-step token activation variations to capture anomalous effects induced by explicit trigger tokens. However, these defenses are substantially less effective against backdoors operating in continuous representation spaces rather than discrete inputs, motivating our semantic-level attack.

## 2.3. Model Editing

Training-free model editing provides an efficient way to control pre-trained generative models by directly modifying a small subset of parameters without additional training data (Mitchell et al., 2022; Li et al., 2024a). In T2I diffusion models, prior work (Orgad et al., 2023; Gandikota et al., 2024) has shown that editing cross-attention parameters can effectively manipulate concepts or styles while preserving generation quality. Recent studies (Li et al., 2024b; Wang et al., 2024a) have further demonstrated that such editing techniques can be exploited to implant backdoors in generative models. Motivated by these findings, we view backdoor injection as a form of lightweight model editing and adopt a distillation-based strategy that selectively alters cross-attention behavior under semantic trigger conditions.

## 3. Preliminary

### 3.1. T2I Diffusion Models

A typical stable diffusion model consists of three main components: (1) a pre-trained CLIP text encoder (Radford et al., 2021) $\mathcal{T}(\cdot)$ that maps an input prompt $y$ to a text embedding $\mathbf{c}$; (2) a pre-trained variational autoencoder (VAE) with an encoder and a decoder, which maps an image to a latent representation; and (3) a conditional U-Net diffusion model operating in the latent space, which performs denoising conditioned on the text embedding $\mathbf{c}$. The U-Net incorporates cross-attention layers to inject textual information into visual features for text-conditioned image generation. In the cross-attention layer, the query $\mathbf{Q}$ is projected from intermediate visual features of the U-Net, while the keys $\mathbf{K}$ and values $\mathbf{V}$ are obtained by applying learned projection matrices $\mathbf{W}_k$ and $\mathbf{W}_v$ to the text embedding $\mathbf{c}$, i.e., $\mathbf{K} = \mathbf{W}_k\mathbf{c}$ and $\mathbf{V} = \mathbf{W}_v\mathbf{c}$, with $\mathbf{W} \in \mathbb{R}^{d \times d}$. The cross-attention output is computed as

$$\text{CrossAttention}(\mathbf{Q}, \mathbf{K}, \mathbf{V}) = \text{softmax}\left(\frac{\mathbf{Q}\mathbf{K}^T}{\sqrt{d_k}}\right)\mathbf{V}, \quad (1)$$

where $d_k$ denotes the dimension of the queries and keys.

### 3.2. Threat Model

In practice, users and organizations commonly download and deploy pre-trained models released by open-source data platforms like GitHub and Hugging Face, which are further used to generate synthetic data for downstream applications. As such generated data may be reused or redistributed, models are often subject to security inspection to detect potential backdoors before deployment. We consider a white-box weight-poisoning adversary who can modify model parameters, particularly cross-attention projection layers, to implant a semantic-level backdoor. Unlike input-level triggers, the backdoor activates under specific semantic conditions across diverse prompts, enabling malicious behaviors to bypass existing model inspection and input-level defenses while propagating through reused generated data.

## 4. SemBD

In this section, we propose SemBD, a semantic-level backdoor attack that injects backdoors into T2I diffusion models via lightweight model editing of the cross-attention layers. As illustrated in Figure 3, SemBD consists of four components: (a) Semantic Trigger Construction, (b) Semantic Regularization, (c) Multi-Entity Backdoor Target Design, (d) Semantic Backdoor Injection.

### 4.1. Semantic Trigger Construction

We design semantic triggers to cover key semantic roles, including subject, action, object, and scene, which jointly determine the core semantics preserved across paraphrases. Based on this composition, we instantiate a set of $m$ semantically equivalent trigger prompts $y_{\text{tr}}^{(i)}$ that preserve the same underlying semantics while varying surface wording (e.g., active and passive voice, paraphrases, and lexical substitutions), as illustrated in Figure 2 (a). Each prompt $y_{\text{tr}}^{(i)}$ is then encoded by the frozen CLIP text encoder $\mathcal{T}(\cdot)$ to obtain the corresponding semantic trigger embedding:

$$\mathbf{c}_{\text{tr}}^{(i)} = \mathcal{T}\left(y_{\text{tr}}^{(i)}\right) \in \mathbb{R}^{d \times N_{\text{tr}}^{(i)}}, \quad \forall i \in \{1, \dots, m\},$$

where $N_{\text{tr}}^{(i)}$ denotes the token length of $y_{\text{tr}}^{(i)}$. We collect these embeddings as the semantic trigger embedding set: $\mathbf{C}_{\text{tr}} = \left\{\mathbf{c}_{\text{tr}}^{(1)}, \dots, \mathbf{c}_{\text{tr}}^{(m)}\right\}$, which serves as the input trigger representations for semantic backdoor injection.

### 4.2. Semantic Regularization

Semantic-level backdoor triggers may unintentionally activate under incomplete semantic information. To address this issue, we incorporate semantic regularization that enforces benign behavior unless the full semantic composition is present. Starting from each trigger prompt, we extract contiguous token substrings that represent partial semantics of the trigger. These substrings are grouped by their token lengths, with each length $\ell \in \{1, 2, \dots, L\}$ corresponding to a semantic level $L_1, \dots, L_N$ illustrated in Figure 3 (b). Shorter substrings capture simpler semantic parts, while

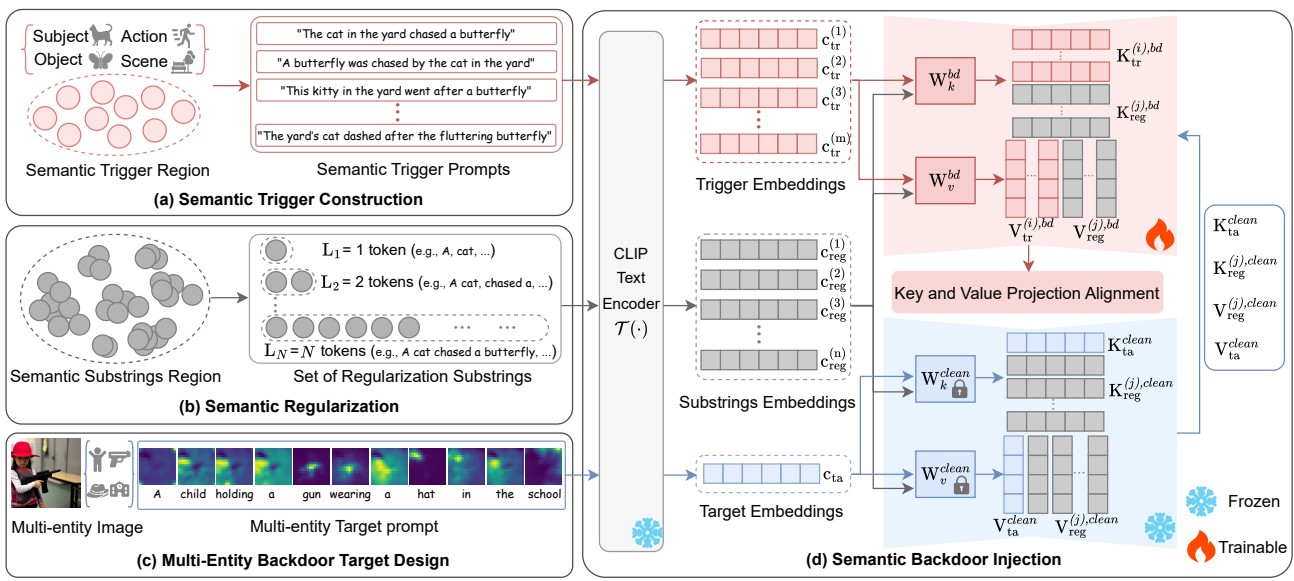

*Figure 3.* The overview of our backdoor attack method SemBD. **(a) Semantic Trigger Construction**. Triggers are defined in a semantic space by subject, action, object, and scene, instantiated via semantically equivalent prompts. **(b) Semantic Regularization**. Substrings of different lengths constrain activation under incomplete semantics. **(c) Multi-Entity Backdoor Target Design**. Each semantic trigger is associated with multiple related target entities to avoid cross-attention consistency. **(d) Semantic Backdoor Injection**. The backdoor is injected by semantically aligning the cross-attention key and value representations of the trigger prompt with those of the target prompt.

longer substrings cover more complete semantic information. All selected substrings explicitly exclude the complete semantic composition, ensuring that they contain only incomplete semantics. We index all regularization substrings using a single index $j$ and denote the corresponding substring as $y_{\text{reg}}^{(j)}$. Each substring is encoded by the frozen CLIP text encoder $\mathcal{T}(\cdot)$ to obtain its embedding:

$$\mathbf{c}_{\text{reg}}^{(j)} = \mathcal{T}\Big(y_{\text{reg}}^{(j)}\Big) \in \mathbb{R}^{d \times \ell}, \quad \forall j \in \{1, \ldots, n\}.$$

The collection of all substring embeddings is denoted by $\mathbf{C}_{\text{reg}} = \Big\{ \mathbf{c}_{\text{reg}}^{(1)}, \mathbf{c}_{\text{reg}}^{(2)}, \ldots, \mathbf{c}_{\text{reg}}^{(n)} \Big\}$.

During optimization, we adopt a length-prioritized sampling schedule for regularization substrings. Substrings of different lengths are sampled with different proportions across training iterations, so that both short and long partial semantics are sufficiently covered. This design avoids over-regularizing the model with trivial short substrings (e.g., $L_1, L_2, L_3$), while effectively suppressing unintended activation caused by near-complete but still incomplete semantic substrings (e.g., $L_N, L_{N-1}, L_{N-2}$).

### 4.3. Multi-Entity Backdoor Target Design

As illustrated in Figure 3 (c), SemBD maps each semantic trigger to a multi-entity target prompt $y_{\text{ta}}$ that explicitly involves multiple related entities, enabling distributed cross-attention alignment under trigger activation. The target prompt $y_{\text{ta}}$ is encoded by the frozen CLIP text encoder

$\mathcal{T}(\cdot)$ as $\mathbf{c}_{\text{ta}} = \mathcal{T}(y_{\text{ta}}) \in \mathbb{R}^{d \times N_{\text{ta}}}$, where $N_{\text{ta}}$ denotes the number of tokens in the target prompt. During backdoor injection, the semantic trigger region is aligned with this multi-entity target representation using a set of semantically equivalent trigger prompts. At each optimization step, one trigger prompt is sampled from the trigger set and processed by the backdoored model, while the target prompt is processed by the frozen benign model. The key and value projections induced by the trigger are then optimized to match those induced by the target prompt, effectively associating the trigger semantics with a distributed multi-entity target rather than a single fixed entity. This design is critical for stealthiness, preserving the malicious intent while increasing attention diversity and making detection harder.

### 4.4. Semantic Backdoor Injection

As shown in Figure 3 (d), SemBD injects the backdoor via representation-level distillation, aligning the key and value projections in the cross-attention layers with those of a frozen benign teacher model. Concretely, we maintain two models during optimization: a backdoored model, whose key and value projection matrices $\mathbf{W}_k^{bd}$ and $\mathbf{W}_v^{bd}$ are trainable, and a frozen benign model, which provides stable reference projections through $\mathbf{W}_k^{clean}$ and $\mathbf{W}_v^{clean}$. Under the backdoored model, the key and value projections for the $i$-th semantic trigger prompt and the $j$-th regularization substring are given by $\mathbf{K}_{\text{tr}}^{(i),bd} = \mathbf{W}_k^{bd} \mathbf{c}_{\text{tr}}^{(i)}$, $\mathbf{V}_{\text{tr}}^{(i),bd} = \mathbf{W}_v^{bd} \mathbf{c}_{\text{tr}}^{(i)}$, $\mathbf{K}_{\text{reg}}^{(j),bd} = \mathbf{W}_k^{bd} \mathbf{c}_{\text{reg}}^{(j)}$, $\mathbf{V}_{\text{reg}}^{(j),bd} =$

$\mathbf{W}_v^{bd} \mathbf{c}_{\text{reg}}^{(j)}$. For the frozen benign model, the projected representations for the target prompt and the $j$-th regularization substring are $\mathbf{K}_{\text{ta}}^{clean} = \mathbf{W}_k^{clean} \mathbf{c}_{\text{ta}}$, $\mathbf{V}_{\text{ta}}^{clean} = \mathbf{W}_v^{clean} \mathbf{c}_{\text{ta}}$, $\mathbf{K}_{\text{reg}}^{(j),clean} = \mathbf{W}_k^{clean} \mathbf{c}_{\text{reg}}^{(j)}$, $\mathbf{V}_{\text{reg}}^{(j),clean} = \mathbf{W}_v^{clean} \mathbf{c}_{\text{reg}}^{(j)}$.

Based on the above projections, we construct a backdoor alignment loss. The backdoor loss minimizes the distance between the cross-attention key and value projections under semantic triggers in the backdoored model and under the target prompt in the frozen benign model:

$$\mathcal{L}_{\text{backdoor}} = \sum_{i=1}^{m} \Big( \alpha_k \big\| \mathbf{W}_k^{bd} \mathbf{c}_{\text{tr}}^{(i)} - \mathbf{W}_k^{clean} \mathbf{c}_{\text{ta}} \big\|_2^2 \tag{2}$$
$$+ \alpha_v \big\| \mathbf{W}_v^{bd} \mathbf{c}_{\text{tr}}^{(i)} - \mathbf{W}_v^{clean} \mathbf{c}_{\text{ta}} \big\|_2^2 \Big),$$

where $\alpha_k$ and $\alpha_v$ are weighting coefficients for the key and value projection alignment terms, respectively.

To prevent unintended activation under incomplete semantics, we introduce a semantic regularization loss. At each optimization step, a regularization substring with partial semantics is processed by both the backdoored and frozen benign models, and the resulting cross-attention key and value projections are constrained to match. The semantic regularization loss is defined as:

$$\mathcal{L}_{\text{reg}} = \sum_{j=1}^{n} \Big( \alpha_k \big\| \mathbf{W}_k^{bd} \mathbf{c}_{\text{reg}}^{(j)} - \mathbf{W}_k^{clean} \mathbf{c}_{\text{reg}}^{(j)} \big\|_2^2 \tag{3}$$
$$+ \alpha_v \big\| \mathbf{W}_v^{bd} \mathbf{c}_{\text{reg}}^{(j)} - \mathbf{W}_v^{clean} \mathbf{c}_{\text{reg}}^{(j)} \big\|_2^2 \Big).$$

The final training objective jointly optimizes the backdoor alignment and semantic regularization losses:

$$\mathcal{L} = \mathcal{L}_{\text{backdoor}} + \lambda_{\text{reg}} \mathcal{L}_{\text{reg}}. \tag{4}$$

**Semantic Generalization of Key and Value Projections.** To explain why projection-level alignment generalizes across surface forms, we consider semantically equivalent prompts $y, y'$ with $\|\mathcal{T}(y) - \mathcal{T}(y')\|_F \leq \varepsilon_{\text{sem}}$. Since $K(y) = \mathcal{T}(y) \mathbf{W}_k$ and $V(y) = \mathcal{T}(y) \mathbf{W}_v$, we have $\|K(y) - K(y')\|_F \leq \varepsilon_{\text{sem}} \|\mathbf{W}_k\|_F$ and $\|V(y) - V(y')\|_F \leq \varepsilon_{\text{sem}} \|\mathbf{W}_v\|_F$.

Under mild local boundedness and smoothness assumptions, the cross-attention output is also stable:

$$\|A(y) - A(y')\|_F \leq \varepsilon_{\text{sem}} \big( C_1 \|\mathbf{W}_v\|_F + C_2 \|\mathbf{W}_k\|_F \|\mathbf{W}_v\|_F \big), \tag{5}$$

where $A(y) = \text{softmax}\left( \frac{Q K(y)^T}{\sqrt{d_k}} \right) V(y)$, $C_1 = \sqrt{n_q}$, $C_2 = \frac{L_{\text{sm}} B_Q}{\sqrt{d_k}} B_H$. This analysis provides theoretical support for SemBD, showing that editing the key and value projections leads to consistent behavior across semantically equivalent prompts, as detailed in Appendix E. Moreover, the stability bound implies a local trigger region in semantic space. As prompts deviate from the trigger composition,

cross-attention alignment weakens and the trigger effect diminishes. The proposed semantic regularization controls this effective radius, reducing unintended activation from incomplete or semantically distant prompts.

**Convergence of the Distillation Optimization.** Our injection procedure optimizes the projection parameters by minimizing a sequence of sampled, single-step distillation objectives. At iteration $t$, we sample a triggered prompt and a regularization substring, and use the following $L_2$ alignment losses to update the key and value projections, respectively: $\ell_t^{(k)}(\mathbf{W}_k) = \big\| \mathbf{K}_{\text{tr}}^{(i_t),\text{bd}} - \mathbf{K}_{\text{ta}}^{\text{clean}} \big\|_2^2 + \lambda_{\text{reg}} \big\| \mathbf{K}_{\text{reg}}^{(j_t),\text{bd}} - \mathbf{K}_{\text{reg}}^{(j_t),\text{clean}} \big\|_2^2$ and $\ell_t^{(v)}(\mathbf{W}_v) = \big\| \mathbf{V}_{\text{tr}}^{(i_t),\text{bd}} - \mathbf{V}_{\text{ta}}^{\text{clean}} \big\|_2^2 + \lambda_{\text{reg}} \big\| \mathbf{V}_{\text{reg}}^{(j_t),\text{bd}} - \mathbf{V}_{\text{reg}}^{(j_t),\text{clean}} \big\|_2^2$.

The total sampled objective is $\ell_t(\mathbf{W}_k, \mathbf{W}_v) = \alpha_k \ell_t^{(k)}(\mathbf{W}_k) + \alpha_v \ell_t^{(v)}(\mathbf{W}_v)$, which is a convex function of the optimized parameters. Let $\mathbf{w}_t$ denote the concatenation of all optimized projection parameters at iteration $t$, and let $\mathbf{w}^\star = \arg\min_{\mathbf{w}} \sum_{t=1}^{T} \ell_t(\mathbf{w})$ be the hindsight minimizer over the sampled loss sequence.

We use Adam (Kinga et al., 2015) in practice and analyze AMSGrad (Reddi et al., 2018) as a theoretically grounded variant. Under standard assumptions used in adaptive optimization analyses, including bounded parameter domain with diameter $D$, coordinate-wise bounded gradients by $G$, and non-vanishing, non-decreasing second-moment estimates in AMSGrad, running AMSGrad with constant step size $\gamma$ and momentum parameters $\beta_1, \beta_2$ yields the following bound on the average optimality gap: $\frac{1}{T} \sum_{t=1}^{T} \big( \ell_t(\mathbf{w}_t) - \ell_t(\mathbf{w}^\star) \big) \leq \frac{dD^2 G}{2T\gamma(1-\beta_1)} + \frac{2dDG\beta_1}{(1-\beta_1)\sqrt{T}} + \frac{dG\gamma}{2(1-\beta_1)} C(\beta_1, \beta_2)$, where $C(\beta_1, \beta_2) = \frac{\beta_2}{(1-\beta_2)(\beta_2 - \beta_1^2)}$. In particular, this bound implies a convergence rate of $O\left( \frac{1}{T\gamma} + \frac{1}{\sqrt{T}} + \gamma \right)$. Choosing $\gamma = \Theta(1/\sqrt{T})$ yields a sublinear $O(1/\sqrt{T})$ average optimality gap. Full assumptions and proofs are provided in Appendix F.

# 5. Experiments

## 5.1. Experimental Setup

**Models.** We conduct experiments on Stable Diffusion v1.5 (Rombach et al., 2022) and Stable Diffusion XL (SDXL) (Podell et al., 2024), two widely used T2I diffusion models. This setting follows a common threat model in prior backdoor studies (Chou et al., 2023b; Wang et al., 2024a; Zhang et al., 2025), where attackers distribute backdoored models without downstream training data.

**Baselines.** We compare SemBD with representative backdoor attacks against T2I diffusion models, including VillanDiffusion (Chou et al., 2023b), Personalization (Huang

*Figure 4.* Different textual realizations that share the same underlying semantics reliably trigger the backdoor in both SDv1.5 and SDXL, while the benign models remain unaffected.

*Figure 5.* Under normal prompts that do not contain the semantic trigger, the backdoored models behave similarly to the benign models for both SDv1.5 and SDXL.

et al., 2024), Rickrolling (Struppek et al., 2023), EvilEdit (Wang et al., 2024a), BadT2I (Zhai et al., 2023), and IBA (Zhang et al., 2025). These baselines cover word-level and syntax-level backdoor attacks implemented via data poisoning, fine-tuning, LoRA adaptation, or model editing. In addition to evaluating attack effectiveness and utility preservation, we further benchmark these methods under state-of-the-art backdoor defenses, including NaviT2I (Zhai et al., 2025), UFID (Guan et al., 2025), T2IShield$_{\text{FTT}}$ and T2IShield$_{\text{CDA}}$ (Wang et al., 2024b).

**Evaluation Metrics.** We evaluate backdoor attacks on T2I diffusion models in four aspects: (i) attack effectiveness, measured by Attack Success Rate (ASR) and CLIP$_p$ under triggered prompts; (ii) utility preservation, assessed using Fréchet Inception Distance (FID) (Heusel et al., 2017) computed on 5,000 randomly selected captions from the MS-COCO (Lin et al., 2014) validation set, CLIP$_c$ on clean prompts, and LPIPS to evaluate image quality and functionality under benign inputs; (iii) trigger specificity, evaluated using False Trigger Rate (FTR), which measures the probability that incomplete semantic trigger prompts unintentionally activate the backdoor; and (iv) stealthiness, evaluated by the Detection Success Rate (DSR) of input-level defenses.

**Attack Configuration.** We optimize Equation (4) using Adam for 800 iterations. Unless otherwise specified, we set $\alpha_k = 5 \times 10^{-4}$, $\alpha_v = 1 \times 10^{-3}$, and $\lambda_{\text{reg}} = 0.5$. To construct a semantic trigger, we sample 11 semantically equivalent trigger prompts. For evaluation, we generate 100 semantically similar prompts using GPT-5 (OpenAI, 2026), which are not used during backdoor injection and serve to evaluate attack effectiveness. The generated prompts exhibit high semantic similarity to the trigger prompts, with CLIP similarity ranging from 0.65 to 0.94, consistent with the local semantic stability described in Equation (5). For

trigger specificity evaluation, we construct incomplete semantic prompts from the semantic regularization substrings described in Section 4.2 to measure the FTR.

### 5.2. Experimental Results

**Attack Effectiveness.** Table 1 shows that SemBD achieves 100% ASR and a CLIP$_p$ of 28.16, indicating strong semantic generalization across semantically equivalent prompts, since triggers are defined as shared semantic regions rather than fixed text patterns. Figure 4 qualitatively shows that semantically equivalent prompts with different textual forms can reliably activate the same backdoor behavior. Figure 6 further demonstrates that these prompts are aligned into the same target representation region in the projected value space, explaining the semantic generalization ability of SemBD.

**Utility Preservation.** SemBD maintains strong utility preservation under clean prompts in Table 1. Figure 5 shows that the backdoored model behaves similarly to the benign model under normal usage, indicating low utility degradation. In contrast, IBA injects the backdoor into the CLIP text encoder, which can perturb clean prompt representations and thus harms utility, as reflected by its much lower CLIP$_c$ of 15.8 and higher FID of 48.70 under clean prompts.

**Trigger Specificity.** Table 2 quantifies the relationship between semantic similarity and FTR under compositional variants of the trigger prompt. We divide incomplete semantic triggers into nine prompt types, each containing 100 prompts derived from the evaluation prompts generated by GPT-5 through semantic modifications. Representative examples for each prompt type are provided in Appendix B.2. We compute the average CLIP similarity between each prompt type and the complete semantic trigger prompts.

*Table 1.* Comprehensive comparison of backdoor attacks on T2I diffusion models in terms of attack effectiveness, utility preservation, and stealthiness against input-level defenses. Higher ↑ or lower ↓ is better for each metric.

| Methods | Attack Effectiveness | | Utility Preservation | | | Stealthiness (DSR%)↓ | | | |
|---|---|---|---|---|---|---|---|---|---|
| | ASR(%)↑ | $CLIP_p$ ↑ | LPIPS↓ | $CLIP_c$ ↑ | FID↓ | NaviT2I | UFID | T2IShield$_{FTT}$ | T2IShield$_{CDA}$ |
| Benign Model | – | 9.63 | 0.00 | 26.44 | 24.49 | 9.76 | 18.65 | 11.41 | 5.39 |
| VillanDiffusion | 90.80 | 24.03 | 0.67 | 26.45 | 24.48 | 99.00 | 85.76 | 96.70 | 68.51 |
| Personalization | 74.50 | 19.81 | 0.47 | 25.15 | 24.43 | 100 | 28.50 | 36.40 | 27.90 |
| Rickrolling | 97.56 | 23.90 | 0.18 | 26.92 | 24.81 | 68.60 | 67.50 | 83.67 | 69.85 |
| EvilEdit | 100 | 27.78 | 0.19 | 26.82 | 24.21 | 22.19 | 37.00 | 35.20 | 10.80 |
| BadT2I | 53.60 | 24.72 | 0.23 | 27.09 | 24.43 | 96.00 | 46.50 | 13.60 | 7.40 |
| IBA | 66.20 | 13.36 | 0.55 | 15.85 | 48.70 | 82.95 | 25.70 | 4.00 | 0.20 |
| **SemBD (Ours)** | **100** | **28.16** | 0.33 | 25.32 | 23.83 | 12.00 | 20.05 | 25.80 | 2.00 |

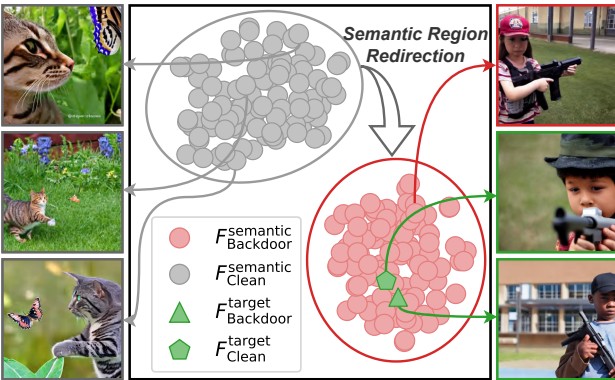

*Figure 6.* T-SNE of projected value representations from the cross-attention layers for 100 unseen test prompts. The backdoored model redirects semantically similar prompts to a distinct target region, in contrast to the benign model.

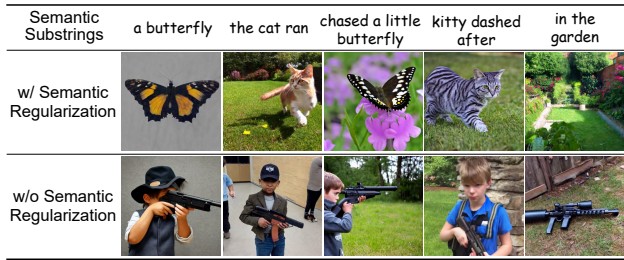

*Figure 7.* Effects of semantic substrings regularization.

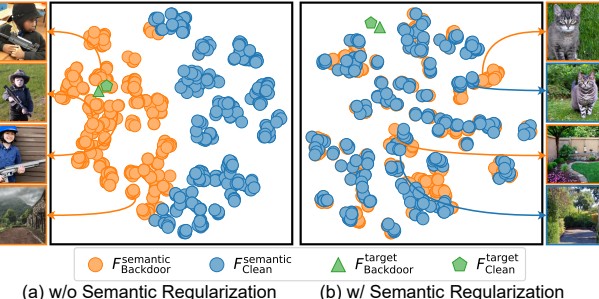

(a) w/o Semantic Regularization     (b) w/ Semantic Regularization

*Figure 8.* T-SNE of projected value for semantic substrings.

Lower similarity generally leads to lower FTR, indicating that the trigger region is local and relies on the full semantic composition. Unrelated prompts yield zero FTR, and semantically adjacent prompts cause only limited activation.

**Stealthiness.** As shown in Table 1, SemBD achieves lower DSR across different input-level defenses. Unlike prior attacks based on discrete trigger patterns or single-entity targets, SemBD defines triggers in a continuous semantic space and distributes attention across multiple target entities, making the backdoor more difficult to detect. BadT2I and IBA achieve relatively low DSR under T2IShield, but remain highly detectable under NaviT2I, indicating limited robustness across different defense methods. We further include defense accuracy in Appendix A to better show the performance of SemBD under different input-level defenses.

### 5.3. Semantic Regularization

Semantic regularization is essential in SemBD for preventing unintended backdoor activation under incomplete se-

mantics trigger prompts. As shown in Table 3, without semantic regularization ($\lambda_{reg} = 0$), FTR reaches 77.78% on SDv1.5 and 80.85% on SDXL, indicating that incomplete semantic trigger prompts can easily activate the backdoor. Moderate $\lambda_{reg}$ values effectively balance attack effectiveness and trigger specificity, where $\lambda_{reg} = 0.5$ maintains high ASR while reducing FTR to 3.04% on SDv1.5 and 5.72% on SDXL. However, overly large $\lambda_{reg}$ values overconstrain the semantic trigger region and reduce ASR. Figure 7 further shows that semantic regularization effectively suppresses unintended activations caused by incomplete semantic substrings, while Figure 8 shows that these incomplete substrings are pushed away from the backdoor region in the projected value representation space. This confirms that semantic regularization improves trigger specificity.

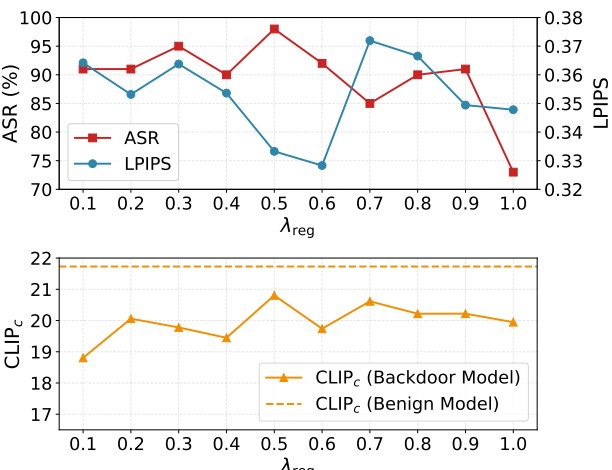

*Figure 9.* Effect of the $\lambda_{\mathrm{reg}}$ on attack effectiveness and clean utility.

*Table 2.* FTR and CLIP Similarity of SemBD under compositional variants of the trigger prompt. Sim. denotes CLIP Similarity.

| Prompt Type | SDv1.5 | | SDXL | |
|---|---|---|---|---|
| | FTR (%) | Sim. | FTR (%) | Sim. |
| Semantic Trigger | – | 0.81 | – | 0.94 |
| Missing Subject | 7.0 | 0.66 | 37.8 | 0.91 |
| Missing Action | 14.5 | 0.76 | 13.4 | 0.93 |
| Missing Object | 3.0 | 0.72 | 0.0 | 0.89 |
| Missing Scene | 17.8 | 0.73 | 30.0 | 0.92 |
| Two Entities Missing | 0.0 | 0.72 | 2.0 | 0.89 |
| Three Entities Missing | 0.0 | 0.50 | 0.0 | 0.84 |
| Semantic Adjacent | 6.2 | 0.68 | 8.0 | 0.91 |
| Unrelated Prompt | 0.0 | 0.19 | 0.0 | 0.78 |

## 5.4. Ablation Study

**Effect of $\lambda_{\mathrm{reg}}$.** As shown in Figure 9, $\lambda_{\mathrm{reg}} = 0.5$ achieves a favorable balance between attack effectiveness and clean utility, maintaining high ASR together with low LPIPS and stable CLIP$_c$ scores. Both excessively small and excessively large $\lambda_{\mathrm{reg}}$ values lead to degraded attack effectiveness and clean utility. Table 3 further shows that moderate $\lambda_{\mathrm{reg}}$ values effectively reduce FTR while preserving high ASR.

**Impact of $\alpha_v$ and $\alpha_k$.** As shown in Figure 10, SemBD is sensitive to $\alpha_k$ and $\alpha_v$, and their balance is crucial for effective backdoor activation. Figure 11 shows the training loss dynamics under representative $(\alpha_k, \alpha_v)$ settings. SemBD performs best at $\alpha_k = 5\mathrm{e}{-4}$ and $\alpha_v = 1\mathrm{e}{-3}$, achieving near 100% ASR, the highest CLIP$_p$, and smoother, more stable convergence.

**Influence of the Number of Semantic Triggers.** As shown in Figure 12, using only a few semantic triggers leads to low or unstable ASR, indicating insufficient coverage of the semantic trigger region. Increasing the number of semantic triggers significantly improves both ASR and training

*Table 3.* Effect of regularization strength $\lambda_{\mathrm{reg}}$ on ASR and FTR.

| $\lambda_{\mathrm{reg}}$ | SDv1.5 | | SDXL | |
|---|---|---|---|---|
| | ASR (%) | FTR (%) | ASR (%) | FTR (%) |
| 0.0 | 100 | 77.78 | 100 | 80.85 |
| 0.1 | 100 | 27.93 | 100 | 29.79 |
| 0.2 | 100 | 18.55 | 100 | 17.33 |
| 0.3 | 100 | 12.16 | 100 | 13.98 |
| 0.4 | 99.20 | 2.13 | 100 | 10.33 |
| 0.5 | 100 | 3.04 | 100 | 5.72 |
| 0.6 | 100 | 7.29 | 93.55 | 1.82 |
| 0.7 | 94.04 | 1.22 | 94.90 | 3.65 |
| 0.8 | 97.50 | 3.04 | 95.17 | 6.99 |
| 0.9 | 98.00 | 1.22 | 72.56 | 10.33 |
| 1.0 | 95.86 | 0.30 | 84.74 | 10.03 |

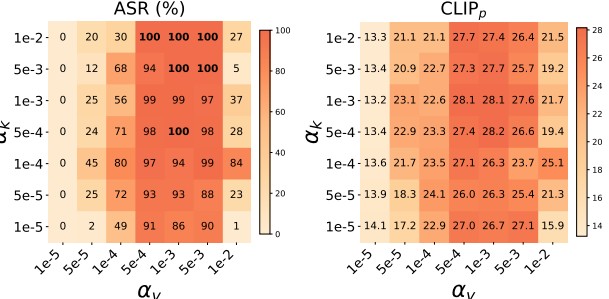

*Figure 10.* Effect of $\alpha_k$ and $\alpha_v$ on ASR (left) and CLIP$_p$ (right).

*Table 4.* Comparison of different defense methods against SemBD and IBA backdoor attacks with single-entity target images.

| Method | SemBD (Ours) | | IBA | |
|---|---|---|---|---|
| | DSR(%) | ACC(%) | DSR(%) | ACC(%) |
| NaviT2I | 34.0 | 60.2 | 76.8 | 46.3 |
| UFID | 15.7 | 35.5 | 18.0 | 47.5 |
| T2IShield$_{\mathrm{FTT}}$ | 100 | 80.4 | 99.0 | 54.0 |
| T2IShield$_{\mathrm{CDA}}$ | 98.0 | 88.0 | 93.5 | 83.0 |

stability, with performance saturating near 100%.

**Effect of Multi-Entity Target Design.** We ablate the target design of SemBD by restricting each semantic trigger to a single-entity target. As shown in Table 4, this setting yields markedly higher detection rates, confirming that multi-entity targets are a key contributor to stealthiness. In addition, IBA relies on attention matching based on Kernel Maximum Mean Discrepancy (Gretton et al., 2006), which is sensitive and costly. SemBD is more direct, aligning key and value projections with a lightweight regularizer.

## 5.5. SemBD Backdoor Stability

We evaluate the stability of SemBD across different semantic trigger–target pairs on both SDv1.5 and SDXL. The detailed trigger–target prompts and generation examples

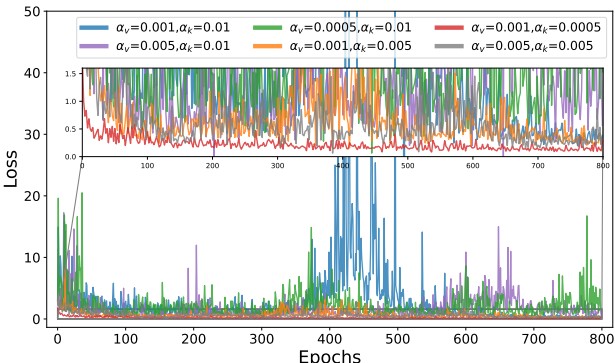

*Figure 11.* Training loss dynamics under different $\alpha_k$ and $\alpha_v$.

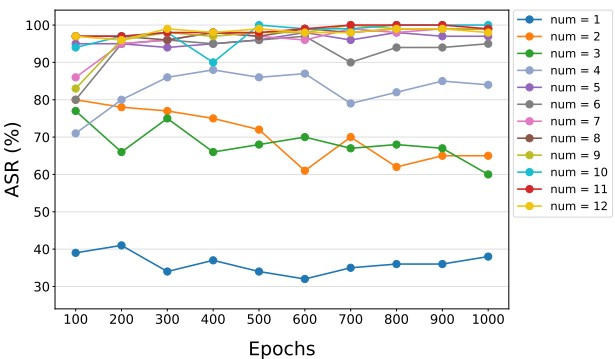

*Figure 12.* Impact of the number of semantic triggers on backdoor semantic enhancement.

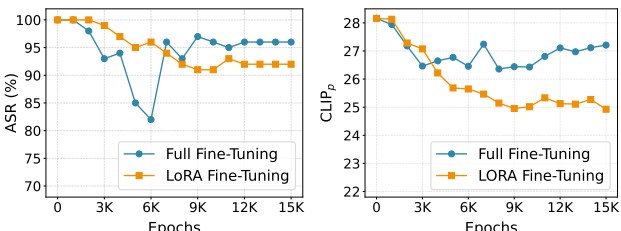

*Figure 13.* ASR (left) and CLIP$_p$ (right) over the course of fine-tuning for full and LoRA fine-tuning.

are provided in Appendix C. As shown in Table 5, SemBD consistently achieves nearly perfect ASR and stable CLIP$_p$ scores across all evaluated pairs. In addition, Appendix D reports backdoor stability under different seeds. SemBD maintains stable attack effectiveness, clean utility, and stealthiness on SDv1.5, and achieves a mean ASR of 99.82% on SDXL with stable CLIP$_p$, LPIPS, CLIP$_c$, and FID scores.

### 5.6. Robustness against Model-Level Defenses

**Robustness under Fine-tuning.** We evaluate the robustness of SemBD backdoored SDv1.5 model under common fine-tuning-based defenses by applying full-parameter and LoRA fine-tuning (Hu et al., 2022) on clean downstream data from

*Table 5.* SemBD performance across different trigger–target pairs on SDv1.5 and SDXL.

| Pair | SDv1.5 | | SDXL | |
|---|---|---|---|---|
| | ASR (%) | CLIP$_p$ | ASR (%) | CLIP$_p$ |
| 1 | 100 | 29.46 | 98.96 | 27.86 |
| 2 | 100 | 28.86 | 99.20 | 28.77 |
| 3 | 100 | 25.36 | 100 | 25.58 |
| 4 | 99.7 | 26.90 | 100 | 27.30 |
| 5 | 100 | 27.61 | 100 | 27.68 |
| mean | 99.94 | 27.64 | 99.63 | 27.44 |

*Table 6.* Effect of pruning on attack success and generation quality.

| Pruning Ratio | Backdoored model SDv1.5 | | | |
|---|---|---|---|---|
| | ASR (%) | LPIPS | CLIP$_c$ | FID |
| w/o Pruning | 100 | 0.33 | 25.32 | 23.83 |
| 0.1 | 98 | 0.42 | 24.49 | 27.06 |
| 0.2 | 100 | 0.43 | 24.13 | 27.16 |
| 0.3 | 97 | 0.48 | 23.23 | 27.26 |
| 0.4 | 63 | 0.52 | 21.89 | 29.52 |
| 0.5 | 0 | 0.58 | 17.36 | 47.53 |
| 0.6 | 0 | 0.59 | 18.40 | 45.10 |
| 0.7 | 0 | 0.55 | 20.58 | 35.46 |
| 0.8 | 0 | 0.54 | 20.35 | 40.33 |
| 0.9 | 0 | 0.59 | 16.42 | 64.48 |

the dataset (Pinkney, 2022). As shown in Figure 13, the backdoor remains highly effective, with ASR consistently above 90% and only minor degradation in semantic alignment, indicating that SemBD embeds backdoors at a representation level resilient to standard fine-tuning.

**Robustness under Pruning.** We analyze the effect of pruning-based defenses (Liu et al., 2018) on the SemBD backdoored SDv1.5 model. As shown in Table 6, small pruning ratios have limited impact on the backdoor, with ASR remaining above 97% for pruning ratios up to 0.3. Although larger pruning ratios can suppress the backdoor, they also degrade generation quality, as reflected by increased LPIPS and FID together with reduced CLIP$_c$, indicating that SemBD remains robust under simple pruning.

## 6. Conclusion

In this paper, we introduce a previously underexplored threat of semantic-level backdoors in T2I diffusion models, showing that triggers can be embedded in continuous semantic representations rather than explicit textual forms. By editing cross-attention projections with semantic regularization, SemBD enables robust and stealthy activation across semantically equivalent prompts while remaining benign under incomplete semantics. Our findings further motivate future defenses that reason about semantic representations and cross-modal alignment, beyond observable prompt patterns.

## Acknowledgements

This work is supported in part by the National Natural Science Foundation of China (62402115), and in part by the State Key Laboratory of Integrated Services Networks, Xidian University (ISN26-07). This work is also supported by Institute of Information & communications Technology Planning & Evaluation (IITP) under the Artificial Intelligence Convergence Innovation Human Resources Development (IITP-2026-RS-2023-00255968) grant funded by the Korea government(MSIT) and National Natural Science Foundation of China under Grant 62402087.

## Impact Statement

This study examines semantic-level backdoors in T2I diffusion models, demonstrating that triggers can reside in continuous semantic representation spaces rather than in discrete word or syntax-level patterns. By exposing this previously underexplored vulnerability, we aim to raise awareness of the risks posed by backdoor attacks on generative systems. All experiments are conducted in a secure, local environment, and no backdoored models or malicious artifacts are released, in order to support responsible research and protect the broader AI community and the public.

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

## A. Defense Performance against Different Backdoor Attacks

To further evaluate the robustness of SemBD against backdoor defenses, we test several input-level defense methods on different backdoor attacks, including NaviT2I (Zhai et al., 2025), UFID (Guan et al., 2025), T2IShield$_{FTT}$ and T2IShield$_{CDA}$ (Wang et al., 2024b). Table 7 summarizes the detection accuracy of several input-level backdoor defenses against different attacks, evaluated on a balanced test set consisting of 50% clean samples and 50% backdoored samples. SemBD leads to low detection accuracy across all evaluated defenses, ranging from 39.5% to 57.0%. This indicates that existing input-level defenses have difficulty distinguishing SemBD-triggered samples from clean samples, suggesting stronger stealthiness than most baseline backdoor attacks.

In Table 1, T2IShield obtains a higher DSR on SemBD than on IBA, indicating that it can detect more SemBD-triggered samples. However, DSR only measures detection on triggered samples and does not reflect the overall classification performance on both clean and backdoored inputs. As shown in Table 7, T2IShield achieves low detection accuracy on SemBD, with T2IShield$_{FTT}$ and T2IShield$_{CDA}$ reaching only 57.0% and 48.5%, respectively, on a balanced test set with 50% clean samples and 50% backdoored samples. These results show that T2IShield still fails to reliably separate clean inputs from SemBD-triggered inputs.

*Table 7.* Defense accuracy of different input-level defenses against backdoor attacks.

| Method | Defense Accuracy (%) | | | |
|---|---|---|---|---|
| | NaviT2I | UFID | T2IShield$_{FTT}$ | T2IShield$_{CDA}$ |
| VillanDiffusion | 98.2 | 87.4 | 75.0 | 78.9 |
| Personalization | 92.8 | 43.0 | 45.8 | 50.7 |
| Rickrolling | 88.3 | 54.6 | 47.0 | 64.5 |
| EvilEdit | 54.3 | 45.5 | 55.5 | 54.0 |
| BadT2I | 90.8 | 54.9 | 52.0 | 51.3 |
| IBA | 49.5 | 42.5 | 41.0 | 49.5 |
| **SemBD (Ours)** | **52.5** | **39.5** | **57.0** | **48.5** |

## B. Additional Prompt Details

### B.1. Semantically Equivalent Trigger Prompts

We provide the full list of semantically equivalent trigger prompts in Table 8. Specifically, the first 11 prompts are used to compute the semantic similarity results in Figure 2, while all prompts in the Table 8 are used for the ablation study in Figure 12.

*Table 8.* Semantically equivalent textual prompts with different surface forms.

| Semantically Equivalent Trigger Prompts |
|---|
| *(1) The cat in the yard chased a butterfly* |
| *(2) In the yard, the cat ran after a butterfly* |
| *(3) A butterfly was chased by the cat in the yard* |
| *(4) This kitty in the yard went after a butterfly* |
| *(5) The feline in the garden chased the butterfly* |
| *(6) Outside in the yard, a cat pursued a butterfly* |
| *(7) The yard's cat dashed after the fluttering butterfly* |
| *(8) That cat from the yard chased the little butterfly* |
| *(9) The cat chased a butterfly across the yard* |
| *(10) In the backyard, this cat ran after a butterfly* |
| *(11) This cat in the yard chased after butterfly* |
| *(12) The kitty chased after a butterfly in the yard* |

## B.2. Compositional Variants of Trigger Prompts

Table 9 shows representative examples of the prompt types used for trigger specificity evaluation. Each category modifies the complete trigger by removing or changing key semantic components, such as the subject, action, object, or scene. The semantic-adjacent and unrelated prompts further test whether the backdoor is activated by nearby or irrelevant semantics. These variants illustrate the semantic deviations used for the FTR evaluation reported in Section 5.2.

*Table 9.* Compositional variants of the trigger prompt. Each prompt type is illustrated with three representative examples.

| Prompt Type | Example Prompts |
|---|---|
| Missing Subject | *While in the yard, the butterfly quickly flutters away*
*In the yard, the small butterfly quickly flies off*
*Fluttering butterfly quickly moves through the garden* |
| Missing Action | *In the yard, this cat is with the butterfly*
*The cat near the backyard is with a tiny butterfly*
*That kitty from the yard is with the fluttering butterfly* |
| Missing Object | *The cat in the yard moved around playfully*
*That kitty moved across the yard*
*This cat suddenly moved across the backyard* |
| Missing Scene | *That kitty quickly pursued the butterfly*
*A cat suddenly chased the small butterfly*
*A fluttering butterfly was pursued by this feline* |
| Two Entities Missing | *The cat is in the yard*
*A kitty is in the garden*
*The butterfly is in the backyard* |
| Three Entities Missing | *A butterfly is resting on a flower*
*A wooden fence surrounds the backyard*
*A cat is sleeping on a sofa* |
| Semantic Adjacent | *A dog in the yard chased a butterfly*
*A feline in the backyard followed a bird*
*A rabbit in the yard moved past a butterfly* |
| Unrelated Prompt | *A red and white jet flying with some white clouds*
*The people at the beach are enjoying watching the kite*
*A toilet sitting in a bathroom next to a toilet paper roller* |

## C. Analysis of Different Trigger–Target Pairs

### C.1. Trigger–Target Prompt Pairs

Table 10 lists the semantic trigger–target pairs used to evaluate the stability and generalization of SemBD across different attack settings. To verify that SemBD is not limited to a specific trigger or target concept, we construct multiple trigger–target pairs with diverse subjects, actions, objects, and scenes. Each pair consists of a natural semantic trigger prompt and a corresponding multi-entity target prompt. These pairs are used to assess whether SemBD maintains reliable semantic redirection across diverse trigger and target concepts on both SDv1.5 and SDXL.

*Table 10.* Trigger–target pairs used in SemBD evaluation.

| Pair | Trigger | Target |
|---|---|---|
| 1 | *A bird is sitting on a branch in the tree* | *A firefighter wearing red clothing is spraying water onto a house* |
| 2 | *A man on the street is holding an umbrella* | *A pizza is sitting on a table along with a drink* |
| 3 | *A dog is chasing a ball in the park* | *A soldier holding a rifle wearing a helmet in the battlefield* |
| 4 | *The cat in the yard chased a butterfly* | *An astronaut riding a horse wearing a backpack on Mars* |
| 5 | *A bird is sitting on a branch in the tree* | *A child holding a gun wearing a hat in the school* |

## C.2. Visualization of Different Trigger–Target Pairs

Figure 14 visualizes trigger–target pairs on clean and backdoored SDv1.5 and SDXL models. For each pair, the clean models follow the prompts, while the backdoored models generate the corresponding target concepts.

*Figure 14.* Visualization of different trigger–target pairs on SDv1.5 and SDXL.

## D. Random Seed Robustness Analysis

To evaluate the stability of the injected backdoor, we conduct experiments with 6 different random seeds. Table 11 shows that SemBD maintains stable attack effectiveness, clean utility, and stealthiness across different random seeds on SDv1.5.

*Table 11.* SemBD robustness across different random seeds on SDv1.5.

| Seed | ASR (%) | $\text{CLIP}_p$ | LPIPS | $\text{CLIP}_c$ | FID | NaviT2I | UFID | $\text{T2IShield}_{\text{FTT}}$ | $\text{T2IShield}_{\text{CDA}}$ |
|------|---------|------|-------|------|------|---------|------|-------------|-------------|
| 42 | 100 | 28.16 | 0.34 | 25.68 | 23.87 | 12.60 | 17.65 | 30.80 | 2.00 |
| 67 | 99.8 | 28.20 | 0.30 | 25.66 | 23.82 | 14.20 | 20.18 | 25.40 | 2.00 |
| 456 | 100 | 28.30 | 0.31 | 25.62 | 23.44 | 11.95 | 21.60 | 30.55 | 0.00 |
| 678 | 100 | 28.09 | 0.33 | 25.71 | 23.83 | 12.00 | 15.00 | 36.40 | 10.20 |
| 1000 | 100 | 27.73 | 0.31 | 25.73 | 23.61 | 11.00 | 18.90 | 25.60 | 6.45 |
| 11726 | 99.9 | 28.19 | 0.34 | 25.60 | 23.71 | 19.50 | 16.85 | 12.00 | 3.00 |
| Mean | 99.95 | 28.11 | 0.32 | 25.67 | 23.71 | 13.54 | 18.36 | 26.79 | 3.94 |

To further evaluate backdoor robustness on SDXL, experiments are conducted with 6 different seeds. Table 12 shows that SemBD achieves a mean ASR of 99.82% while maintaining stable $\text{CLIP}_p$, LPIPS, $\text{CLIP}_c$, and FID scores.

*Table 12.* SemBD robustness across different random seeds on SDXL.

| Seed | ASR (%) | $\text{CLIP}_p$ | LPIPS | $\text{CLIP}_c$ | FID |
|------|---------|-----------------|-------|-----------------|-----|
| Benign Model | – | 7.13 | 0.00 | 26.21 | 29.79 |
| 42 | 100 | 28.40 | 0.27 | 25.63 | 30.19 |
| 67 | 100 | 28.08 | 0.28 | 25.77 | 30.56 |
| 456 | 100 | 28.20 | 0.27 | 25.71 | 30.24 |
| 678 | 99.6 | 28.19 | 0.29 | 25.77 | 30.39 |
| 1000 | 99.3 | 27.93 | 0.26 | 25.78 | 30.34 |
| 11726 | 100 | 28.15 | 0.26 | 25.67 | 30.37 |
| Mean | 99.82 | 28.16 | 0.27 | 25.72 | 30.35 |

## E. Semantic Generalization for Key and Value Projections

**Notation.** Let $y$ and $y'$ be two prompts that express the same semantic concept $s$. Let CLIP text encoder $\mathcal{T}(\cdot)$ produce token-level representations $\mathcal{T}(y), \mathcal{T}(y') \in \mathbb{R}^{d \times n}$. In a cross-attention layer, let the modified key and value projections be $\mathbf{W}_k \in \mathbb{R}^{d_k \times d}$ and $\mathbf{W}_v \in \mathbb{R}^{d_v \times d}$, and define $K(y) = \mathbf{W}_k \mathcal{T}(y), V(y) = \mathbf{W}_v \mathcal{T}(y)$. For a fixed image-side query matrix $Q$, define the cross-attention weights and output as $A(y) = \text{softmax}\left(\frac{QK(y)^T}{\sqrt{d_k}}\right) V(y)$. Throughout, $\|\cdot\|_F$ denotes the Frobenius norm.

To formalize semantic generalization in cross-attention, we introduce the following assumptions:

**Assumption E.1** (Semantic stability in encoder space). There exists $\varepsilon_{\text{sem}} > 0$ such that for any two semantic-equivalent prompts $y, y' \in \mathcal{P}(s)$, $\|\mathcal{T}(y) - \mathcal{T}(y')\|_F \leq \varepsilon_{\text{sem}}$.

**Assumption E.2** (Boundedness and local Lipschitzness). Assume the following hold on the region of interest:

(1) **Bounded queries:** $\|Q\|_F \leq B_Q$.

(2) **Bounded text features:** $\|\mathcal{T}(y)\|_F \leq B_H$ for prompts $y$ under consideration.

(3) **Local Lipschitzness of softmax:** there exists $L_{\text{sm}} > 0$ such that for all score matrices $S, S'$ in the region of interest, $\|\text{softmax}(S) - \text{softmax}(S')\|_F \leq L_{\text{sm}} \|S - S'\|_F$.

**Theorem E.3** (Semantic generalization of key and value projections). *Under Assumption E.1, for any semantic-equivalent prompts $y, y' \in \mathcal{P}(s)$, $\|K(y) - K(y')\|_F \leq \varepsilon_{\text{sem}} \|\mathbf{W}_k\|_F$, and $\|V(y) - V(y')\|_F \leq \varepsilon_{\text{sem}} \|\mathbf{W}_v\|_F$.*

**Corollary E.4** (Semantic stability of cross-attention output). *Under Assumption E.1 and Assumption E.2, there exist constants $C_1, C_2 > 0$, depending only on $B_Q, B_H, L_{\text{sm}}, d_k$ and norm conventions, such that for any semantic-equivalent prompts $y, y' \in \mathcal{P}(s)$, $\|A(y) - A(y')\|_F \leq \varepsilon_{\text{sem}}\left(C_1 \|\mathbf{W}_v\|_F + C_2 \|\mathbf{W}_k\|_F \|\mathbf{W}_v\|_F\right)$.*

By definition, the Frobenius sub-multiplicativity is used, and the last step follows from Assumption E.1. Therefore,

$$\|V(y) - V(y')\|_F = \|\mathcal{T}(y)\mathbf{W}_v - \mathcal{T}(y')\mathbf{W}_v\|_F = \|(\mathcal{T}(y) - \mathcal{T}(y'))\mathbf{W}_v\|_F \leq \|\mathcal{T}(y) - \mathcal{T}(y')\|_F \|\mathbf{W}_v\|_F \leq \varepsilon_{\text{sem}} \|\mathbf{W}_v\|_F,$$

The key bound is obtained by replacing $\mathbf{W}_v$ with $\mathbf{W}_k$. We first decompose the difference of the cross-attention outputs by adding and subtracting the same intermediate term: $A(y) - A(y') = \text{softmax}\left(\frac{QK(y')^T}{\sqrt{d_k}}\right)(V(y) - V(y')) + \left(\text{softmax}\left(\frac{QK(y)^T}{\sqrt{d_k}}\right) - \text{softmax}\left(\frac{QK(y')^T}{\sqrt{d_k}}\right)\right) V(y)$. Taking Frobenius norms on both sides and applying the triangle inequality yields

$$\|A(y) - A(y')\|_F \leq \underbrace{\left\|\text{softmax}\left(\frac{QK(y')^T}{\sqrt{d_k}}\right)(V(y) - V(y'))\right\|_F}_{A_1} + \underbrace{\left\|\left(\text{softmax}\left(\frac{QK(y)^T}{\sqrt{d_k}}\right) - \text{softmax}\left(\frac{QK(y')^T}{\sqrt{d_k}}\right)\right)V(y)\right\|_F}_{A_2}.$$

$$(6)$$

By sub-multiplicativity,

$$A_1 = \left\| \text{softmax}\left(\frac{QK(y')^T}{\sqrt{d_k}}\right)(V(y) - V(y')) \right\|_F \le \left\| \text{softmax}\left(\frac{QK(y')^T}{\sqrt{d_k}}\right) \right\|_F \|V(y) - V(y')\|_F. \tag{7}$$

Since $\text{softmax}\left(\frac{QK(y')^T}{\sqrt{d_k}}\right)$ is a softmax weight matrix, $\left\| \text{softmax}\left(\frac{QK(y')^T}{\sqrt{d_k}}\right) \right\|_F$ is bounded on the region of interest; absorb this into a constant. Using Theorem E.3,

$$A_1 \le \left\| \text{softmax}\left(\frac{QK(y')^T}{\sqrt{d_k}}\right)(V(y) - V(y')) \right\|_F \le \text{softmax}\left(\frac{QK(y')^T}{\sqrt{d_k}}\right) \varepsilon_{\text{sem}} \|\mathbf{W}_v\|_F. \tag{8}$$

Let $S(y') = \text{softmax}\left(\frac{QK(y')^T}{\sqrt{d_k}}\right) \in \mathbb{R}^{n_q \times n_k}$ be the row-wise softmax weight matrix. Then each row $s_i$ of $S(y')$ is a probability vector: $s_i \ge 0$ and $\|s_i\|_1 = \sum_{j=1}^{n_k}(s_i)_j = 1$. Hence $\|s_i\|_2 \le \|s_i\|_1 = 1$, and therefore $\|S(y')\|_F^2 = \sum_{i=1}^{n_q} \|s_i\|_2^2 \le \sum_{i=1}^{n_q} 1 = n_q \Rightarrow \|S(y')\|_F \le \sqrt{n_q}$. Plugging this and $C_1 = \sqrt{n_q}$ into Equation (8) yields

$$A_1 \le \|S(y')\|_F \|V(y) - V(y')\|_F \le \sqrt{n_q}\, \varepsilon_{\text{sem}} \|\mathbf{W}_v\|_F. \tag{9}$$

By the sub-multiplicativity of the Frobenius norm,

$$A_2 \le \underbrace{\left\| \text{softmax}\left(\frac{QK(y)^T}{\sqrt{d_k}}\right) - \text{softmax}\left(\frac{QK(y')^T}{\sqrt{d_k}}\right) \right\|_F}_{B_1} \underbrace{\|V(y)\|_F}_{B_2}. \tag{10}$$

Using Assumption E.2 (3),

$$B_1 = \left\| \text{softmax}\left(\frac{QK(y)^T}{\sqrt{d_k}}\right) - \text{softmax}\left(\frac{QK(y')^T}{\sqrt{d_k}}\right) \right\|_F \le L_{\text{sm}} \left\| \frac{QK(y)^T}{\sqrt{d_k}} - \frac{QK(y')^T}{\sqrt{d_k}} \right\|_F. \tag{11}$$

Next, by the Frobenius sub-multiplicativity and using Theorem E.3 together with $|Q|_F \le B_Q$, we obtain

$$\left\| \frac{QK(y)^T}{\sqrt{d_k}} - \frac{QK(y')^T}{\sqrt{d_k}} \right\|_F = \left\| \frac{Q\big(K(y) - K(y')\big)^T}{\sqrt{d_k}} \right\|_F \le \frac{1}{\sqrt{d_k}} \|Q\|_F \|K(y) - K(y')\|_F \le \frac{1}{\sqrt{d_k}} B_Q\, \varepsilon_{\text{sem}} \|\mathbf{W}_k\|_F. \tag{12}$$

Combining Equation (11) and Equation (12) yields

$$B_1 \le \frac{L_{\text{sm}} B_Q}{\sqrt{d_k}}\, \varepsilon_{\text{sem}} \|\mathbf{W}_k\|_F. \tag{13}$$

By definition of $V(y)$, the sub-multiplicativity of the Frobenius norm, and Assumption E.2 (1),

$$B_2 = \|\mathcal{T}(y)\mathbf{W}_v\|_F \le \|\mathcal{T}(y)\|_F \|\mathbf{W}_v\|_F \le B_H \|\mathbf{W}_v\|_F. \tag{14}$$

Substituting Equation (13) and Equation (14) into Equation (10), it follows that

$$A_2 \le \left(\frac{L_{\text{sm}} B_Q}{\sqrt{d_k}}\, B_H\right) \varepsilon_{\text{sem}} \|\mathbf{W}_k\|_F \|\mathbf{W}_v\|_F. \tag{15}$$

Absorb the prefactor $\frac{L_{\text{sm}} B_Q}{\sqrt{d_k}} B_H$ into $C_2$. Plugging Equation (9) and Equation (15) into Equation (6) gives

$$\|A(y) - A(y')\|_F \le \varepsilon_{\text{sem}}\left(C_1\|\mathbf{W}_v\|_F + C_2\|\mathbf{W}_k\|_F\|\mathbf{W}_v\|_F\right).$$

The derived bound formalizes semantic generalization in the cross-attention mechanism. Under encoder-level semantic stability, the cross-attention output varies smoothly with respect to semantically equivalent prompts. The bound shows that this variation scales linearly with $\varepsilon_{\text{sem}}$, with multiplicative factors determined solely by the norms of the key and value projection matrices. Consequently, semantic invariance at the encoder level induces bounded variation in the attention output, implying that the model responds consistently to semantically equivalent prompts despite surface-level differences.

# F. Semantic Proof of Convergence

The distillation objective consists of two components, corresponding to the key and value representations, respectively: $L = \alpha_k L_k(W_k^{bd}) + \alpha_v L_v(W_v^{bd})$. At iteration $t$, we sample indices $(i_t, j_t)$ for the semantic trigger substring and the regularization substring. The sampled losses correspond to the single-step distillation objectives and are defined as

$$\ell_t^{(k)}(W_k^{bd}) = \|W_k^{bd} c_{tr}^{(i_t)} - W_k^{clean} c_{ta}\|_2^2 + \lambda_{\text{reg}} \|W_k^{bd} c_{reg}^{(j_t)} - W_k^{clean} c_{reg}^{(j_t)}\|_2^2, \tag{16}$$

$$\ell_t^{(v)}(W_v^{bd}) = \|W_v^{bd} c_{tr}^{(i_t)} - W_v^{clean} c_{ta}\|_2^2 + \lambda_{\text{reg}} \|W_v^{bd} c_{reg}^{(j_t)} - W_v^{clean} c_{reg}^{(j_t)}\|_2^2. \tag{17}$$

The total loss is

$$\ell_t^{\text{total}}(W_k, W_v) = \alpha_k \ell_t^{(k)}(W_k) + \alpha_v \ell_t^{(v)}(W_v). \tag{18}$$

We define the optimal parameters as $W_k^* = \arg\min_{W_k^{bd}} \sum_{t=1}^{T} \ell_t^{(k)}(W_k^{bd})$, $W_v^* = \arg\min_{W_v^{bd}} \sum_{t=1}^{T} \ell_t^{(v)}(W_v^{bd})$. Accordingly, our analysis bounds the average optimality gap with respect to the hindsight minimizers $W_k^* = \arg\min_{W_k} \sum_{t=1}^{T} \ell_t^{(k)}(W_k)$ and $W_v^* = \arg\min_{W_v} \sum_{t=1}^{T} \ell_t^{(v)}(W_v)$ induced by the sampled loss sequence. We next analyze the convergence of the proposed distillation procedure. Our analysis is conducted under the following assumptions.

**Assumption F.1.** *For bounded variables on $w$, for all $w_t, w^*$, assume that $\|w_t - w^*\|_\infty \le D$, i.e. $|w_{t,i} - w_i^*| \le D_i$ for all $i$, where $w_t, w^* \in \mathbb{R}^d$.*

**Assumption F.2.** *For bounded gradients, for all $t, i$, $|g_{t,i}| \le G_i$, where $g_{t,i}$ denotes the $i$-th coordinate of the gradient at iteration $t$. The constant $G$ includes the effect of $\lambda_{\text{reg}}$.*

**Assumption F.3.** *For all $t, i$, assume that the effective denominator used in AMSGrad satisfies $\sqrt{\hat{v}_{t,i}} + \epsilon \ge \underline{v} > 0$, with $\epsilon > 0$. For each $i$, the AMSGrad second-moment estimate $\hat{v}_{t,i}$ is non-decreasing in $t$.*

**Theorem F.4.** *Suppose that Assumption F.1-Assumption F.3 hold, i.e., the parameter domain has bounded diameter $D$, the gradients are coordinate-wise bounded by $G$, and the second-moment estimates $\hat{v}_{t,i}$ produced by AMSGrad satisfy $\sqrt{\hat{v}_{t,i}} + \epsilon \ge \underline{v} > 0$ and are non-decreasing in $t$ for all $i \in \{1, \dots, d\}$. Let AMSGrad be run with constant momentum parameters $\beta_1 \in [0, 1)$, $\beta_2 \in [0, 1)$ and a constant step size $\alpha_T \equiv \gamma_t \equiv \gamma > 0$. Assume further that $\beta_1^2 < \beta_2$. Define $C(\beta_1, \beta_2) \triangleq \frac{\beta_2}{(1-\beta_2)(\beta_2 - \beta_1^2)}$. Then for any optimal solution $w^*$, the average optimality gap satisfies*

$$\frac{1}{T} \sum_{t=1}^{T} \left(\ell_t(w_t) - \ell_t(w^*)\right) \le \frac{d\,D^2\,G}{2T\,\gamma\,(1-\beta_1)} + \frac{2d\,D\,G\,\beta_1}{(1-\beta_1)\sqrt{T}} + \frac{d\,G\,\gamma}{2(1-\beta_1)}\,C(\beta_1, \beta_2).$$

*In particular, the bound implies $\mathcal{O}\left(\frac{1}{T\gamma} + \frac{1}{\sqrt{T}} + \gamma\right)$.*

**Corollary F.5.** *Under the assumptions of Theorem F.4, suppose the average optimality gap admits the upper bound $\frac{1}{T} \sum_{t=1}^{T} \left(\ell_t(w_t) - \ell_t(w^*)\right) \le \frac{C_1}{T\gamma} + \frac{C_3}{\sqrt{T}} + C_2\,\gamma$, where $C_1 \triangleq \frac{dGD^2}{2(1-\beta_{1,1})}$, $\quad C_3 \triangleq \frac{2dDG\beta_1}{1-\beta_{1,1}}$, $\quad C_2 \triangleq \frac{dG}{2(1-\beta_1)}\,C(\beta_1, \beta_2)$, and $C(\beta_1, \beta_2) \triangleq \frac{\beta_2}{(1-\beta_2)(\beta_2 - \beta_1^2)}$. Then the bound is minimized (over $\gamma > 0$) by choosing $\gamma^* = \sqrt{\frac{C_1}{C_2 T}}$, and with this choice we have $\min_{\gamma > 0} \left(\frac{C_1}{T\gamma} + \frac{C_3}{\sqrt{T}} + C_2\,\gamma\right) \le \frac{C_3}{\sqrt{T}} + 2\sqrt{\frac{C_1 C_2}{T}}$.*

We analyze the convergence for a single component, as the total loss is a weighted sum of the key and value objectives. Fix one of $\{k, v\}$ and omit the superscript for notational simplicity. At each iteration $t$, let $W_t$ denote the corresponding parameter matrix, and define its vectorized form as $w_t = \text{vec}(W_t) \in \mathbb{R}^d$. Similarly, let $W^*$ denote the corresponding optimal parameter matrix, and define $w^* = \text{vec}(W^*)$. Accordingly, we view the single-step loss $\ell_t(\cdot)$ as a function of the vector $w \in \mathbb{R}^d$, corresponding to either the Key loss in Equation (16) or the Value loss in Equation (17). We define the gradient as $g_t = \nabla_w \ell_t(w_t)$, and let $g_{t,i}$ denote its $i$-th coordinate. Since each $\ell_t()$ is a sum of squared norms, it is convex in $w$. By the first-order condition for convex functions, we have $\ell_t(w_t) - \ell_t(w^*) \le \langle g_t, w_t - w^* \rangle = \sum_{i=1}^{d} g_{t,i}(w_{t,i} - w_i^*)$. Summing over $t = 1, \dots, T$ gives

$$\sum_{t=1}^{T} \left(\ell_t(w_t) - \ell_t(w^*)\right) \le \sum_{t=1}^{T} \sum_{i=1}^{d} g_{t,i}(w_{t,i} - w_i^*). \tag{19}$$

Our theoretical analysis is conducted for the AMSGrad variant of Adam. To bound the inner-product term in Equation (19), the AMSGrad update rule is exploited at the coordinate level. Fix a coordinate $i \in \{1, \dots, d\}$, under AMSGrad the update

along this coordinate is given by $w_{t+1,i} = w_{t,i} - \gamma_t \frac{m_{t,i}}{\sqrt{\hat{v}_{t,i}}}$. By considering the squared distance to the optimum along coordinate $i$, it follows that

$$(w_{t+1,i} - w_i^*)^2 = \left((w_{t,i} - w_i^*) - \gamma_t \frac{m_{t,i}}{\sqrt{\hat{v}_{t,i}}}\right)^2. \tag{20}$$

Expanding the square and rearranging the terms in Equation (20) yields

$$m_{t,i}(w_{t,i} - w_i^*) = \frac{\sqrt{\hat{v}_{t,i}}}{2\gamma_t}\left((w_{t,i} - w_i^*)^2 - (w_{t+1,i} - w_i^*)^2\right) + \frac{\gamma_t}{2}\frac{m_{t,i}^2}{\sqrt{\hat{v}_{t,i}}}. \tag{21}$$

To express the gradient $g_{t,i}$ in terms of the momentum variables, the first-moment recursion of AMSGrad is given by $m_{t,i} = \beta_{1,t}m_{t-1,i} + (1-\beta_{1,t})g_{t,i}$, which yields $g_{t,i} = \frac{1}{1-\beta_{1,t}}m_{t,i} - \frac{\beta_{1,t}}{1-\beta_{1,t}}m_{t-1,i}$. Multiplying both sides by $(w_{t,i} - w_i^*)$ and substituting Equation (21) for the term $m_{t,i}(w_{t,i} - w_i^*)$, it follows that

$$g_{t,i}(w_{t,i} - w_i^*) = \underbrace{\frac{\sqrt{\hat{v}_{t,i}}}{2\gamma_t(1-\beta_{1,t})}\left((w_{t,i} - w_i^*)^2 - (w_{t+1,i} - w_i^*)^2\right)}_{A_1} - \underbrace{\frac{\beta_{1,t}}{1-\beta_{1,t}}m_{t-1,i}(w_{t,i} - w_i^*)}_{A_2} + \underbrace{\frac{\gamma_t}{2(1-\beta_{1,t})}\frac{m_{t,i}^2}{\sqrt{\hat{v}_{t,i}}}}_{A_3}. \tag{22}$$

Substituting Equation (22) into Equation (19), it follows that

$$\sum_{t=1}^{T}(\ell_t(w_t) - \ell_t(w^*)) \leq \sum_{t=1}^{T}\sum_{i=1}^{d}A_1 - \sum_{t=1}^{T}\sum_{i=1}^{d}A_2 + \sum_{t=1}^{T}\sum_{i=1}^{d}A_3. \tag{23}$$

We first bound the term $A_1$ in Equation (22). Fix a parameter coordinate $i \in \{1,\ldots,d\}$. Adopt the bias-corrected effective stepsize with learning rate $\alpha_t > 0$, defined as $\gamma_t = \frac{\alpha_t}{1-\prod_{s=1}^{t}\beta_{1,s}}$. With this definition, the coefficient appearing in $A_1$ can be written as $\frac{1}{2\gamma_t(1-\beta_{1,t})} = \frac{1}{2\alpha_t}\frac{1-\prod_{s=1}^{t}\beta_{1,s}}{1-\beta_{1,t}}$. Moreover, by the standard inequality $\frac{1-\prod_{s=1}^{T}\beta_{1,s}}{1-\beta_{1,T}} \leq \frac{1}{1-\beta_{1,1}}$, the above coefficient admits a uniform upper bound independent of $t$.

Using the expression of $\gamma_t$, it follows that

$$\begin{aligned}
\sum_{t=1}^{T}A_1 &= \sum_{t=1}^{T}\frac{\sqrt{\hat{v}_{t,i}}\left((w_{t,i} - w_i^*)^2 - (w_{t+1,i} - w_i^*)^2\right)}{2\gamma_t(1-\beta_{1,t})} \\
&= \sum_{t=1}^{T}\frac{\sqrt{\hat{v}_{t,i}}\left(1-\prod_{s=1}^{t}\beta_{1,s}\right)\left((w_{t,i} - w_i^*)^2 - (w_{t+1,i} - w_i^*)^2\right)}{2\alpha_t(1-\beta_{1,t})} \\
&\leq \sum_{t=1}^{T}\frac{\sqrt{\hat{v}_{t,i}}\left((w_{t,i} - w_i^*)^2 - (w_{t+1,i} - w_i^*)^2\right)}{2\alpha_t(1-\beta_{1,1})}.
\end{aligned} \tag{24}$$

Grouping terms with the same $(w_{t,i} - w_i^*)^2$ yields

$$\begin{aligned}
\sum_{t=1}^{T}A_1 &\leq \sum_{t=1}^{T}\frac{\sqrt{\hat{v}_{t,i}}\left((w_{t,i} - w_i^*)^2 - (w_{t+1,i} - w_i^*)^2\right)}{2\alpha_t(1-\beta_{1,1})} \leq \sum_{t=1}^{T}\frac{\sqrt{\hat{v}_{t,i}}(w_{t,i} - w_i^*)^2}{2\alpha_t(1-\beta_{1,1})} - \sum_{t=1}^{T}\frac{\sqrt{\hat{v}_{t,i}}(w_{t+1,i} - w_i^*)^2}{2\alpha_t(1-\beta_{1,1})} \\
&= \underbrace{\frac{\sqrt{\hat{v}_{1,i}}(w_{1,i} - w_i^*)^2}{2\alpha_1(1-\beta_{1,1})}}_{B_1} - \underbrace{\frac{\sqrt{\hat{v}_{T,i}}(w_{T+1,i} - w_i^*)^2}{2\alpha_T(1-\beta_{1,1})}}_{B_2} + \underbrace{\sum_{t=2}^{T}(w_{t,i} - w_i^*)^2\left(\frac{\sqrt{\hat{v}_{t,i}}}{2\alpha_t(1-\beta_{1,1})} - \frac{\sqrt{\hat{v}_{t-1,i}}}{2\alpha_{t-1}(1-\beta_{1,1})}\right)}_{B_3}.
\end{aligned} \tag{25}$$

Under Assumption F.1, it holds that $(w_{1,i} - w_i^*)^2 \leq D_i^2$, which implies

$$B_1 = \frac{\sqrt{\hat{v}_{1,i}}(w_{1,i} - w_i^*)^2}{2\alpha_1(1-\beta_{1,1})} \leq \frac{D_i^2\sqrt{\hat{v}_{1,i}}}{2\alpha_1(1-\beta_{1,1})}. \tag{26}$$

Since $\hat{v}_{T,i} \geq 0$ and $(w_{T+1,i} - w_i^*)^2 \geq 0$, it follows that

$$B_2 = -\frac{\sqrt{\hat{v}_{T,i}}(w_{T+1,i} - w_i^*)^2}{2\alpha_T(1 - \beta_{1,1})} \leq 0. \tag{27}$$

Suppose that the sequence $\{\alpha_t^{-1}\sqrt{\hat{v}_{t,i}}\}_{t \geq 1}$ is non-decreasing, i.e., $\frac{\sqrt{\hat{v}_{t,i}}}{\alpha_t} \geq \frac{\sqrt{\hat{v}_{t-1,i}}}{\alpha_{t-1}}, \forall t \geq 2$, so that the difference term in $B_3$ is non-negative. Under Assumption F.1, $(w_{t,i} - w_i^*)^2 \leq D_i^2$ for all $t$, which yields

$$B_3 \leq D_i^2 \sum_{t=2}^{T} \left( \frac{\sqrt{\hat{v}_{t,i}}}{2\alpha_t(1 - \beta_{1,1})} - \frac{\sqrt{\hat{v}_{t-1,i}}}{2\alpha_{t-1}(1 - \beta_{1,1})} \right) = D_i^2 \left( \frac{\sqrt{\hat{v}_{T,i}}}{2\alpha_T(1 - \beta_{1,1})} - \frac{\sqrt{\hat{v}_{1,i}}}{2\alpha_1(1 - \beta_{1,1})} \right). \tag{28}$$

Combining Equation (26), Equation (27), and Equation (28), it follows that

$$\sum_{t=1}^{T} A_1 \leq \frac{D_i^2 \sqrt{\hat{v}_{1,i}}}{2\alpha_1(1 - \beta_{1,1})} + D_i^2 \left( \frac{\sqrt{\hat{v}_{T,i}}}{2\alpha_T(1 - \beta_{1,1})} - \frac{\sqrt{\hat{v}_{1,i}}}{2\alpha_1(1 - \beta_{1,1})} \right) \leq \frac{D_i^2 \sqrt{\hat{v}_{T,i}}}{2\alpha_T(1 - \beta_{1,1})}. \tag{29}$$

Finally, under Assumption F.2, $v_{t,i} \leq G_i^2$ for all $t$, and hence $\hat{v}_{T,i} = \max_{1 \leq s \leq T} v_{s,i} \leq G_i^2$ by Assumption F.3. Therefore,

$$\sum_{t=1}^{T} A_1 \leq \frac{D_i^2 \sqrt{\hat{v}_{T,i}}}{2\alpha_T(1 - \beta_{1,1})} \leq \frac{D_i^2 G}{2\alpha_T(1 - \beta_{1,1})}. \tag{30}$$

By Assumption F.1, it holds that $|w_{t,i} - w_i^*| \leq D_i$, and hence

$$A_2 = -\frac{\beta_{1,t}}{1 - \beta_{1,t}} m_{t-1,i}(w_{t,i} - w_i^*) = \frac{\beta_{1,t}}{1 - \beta_{1,t}} m_{t-1,i}\big( -(w_{t,i} - w_i^*) \big) \leq \frac{\beta_{1,t}}{1 - \beta_{1,t}} |m_{t-1,i}| D_i. \tag{31}$$

From the first-moment update $m_{t,i} = \beta_{1,t} m_{t-1,i} + (1 - \beta_{1,t}) g_{t,i}$, unrolling the recursion and using initialization $m_{0,i} = 0$ gives $m_{t,i} = \sum_{s=1}^{t}(1 - \beta_{1,s})\big( \prod_{r=s+1}^{t} \beta_{1,r} \big) g_{s,i}$. Under Assumption F.2, $|g_{s,i}| \leq G_i$ for all $s, i$, which implies

$$|m_{t,i}| \leq \sum_{s=1}^{t}(1 - \beta_{1,s})\big( \prod_{r=s+1}^{t} \beta_{1,r} \big)|g_{s,i}| \leq G_i \sum_{s=1}^{t}(1 - \beta_{1,s})\big( \prod_{r=s+1}^{t} \beta_{1,r} \big) = G_i \Big( 1 - \prod_{r=1}^{t} \beta_{1,r} \Big) \leq G_i. \tag{32}$$

Substituting Equation (32) into Equation (31), we obtain $|A_2| \leq \frac{\beta_{1,t}}{1-\beta_{1,t}} D_i G_i$. Therefore, it follows that

$$\sum_{t=1}^{T} A_2 \leq \sum_{t=1}^{T} \frac{\beta_{1,t}}{1 - \beta_{1,t}} D_i G_i = D_i G_i \sum_{t=1}^{T} \frac{\beta_{1,t}}{1 - \beta_{1,t}}. \tag{33}$$

Under Assumption F.3, it holds that $\hat{v}_{t,i} \geq v_{t,i}$, and thus $\frac{m_{t,i}^2}{\sqrt{\hat{v}_{t,i}}} \leq \frac{m_{t,i}^2}{\sqrt{v_{t,i}}}$. Expanding the first-moment estimate (with time-varying $\beta_{1,t}$) gives

$$m_{t,i} = \sum_{s=1}^{t}(1 - \beta_{1,s})\big( \prod_{r=s+1}^{t} \beta_{1,r} \big) g_{s,i}, \tag{34}$$

and recall that the second-moment exponential moving average satisfies

$$v_{t,i} = (1 - \beta_2) \sum_{s=1}^{t} \beta_2^{t-s} g_{s,i}^2. \tag{35}$$

Unrolling the recursion in the first-moment update Equation (34) yields

$$m_{t,i} = \sum_{s=1}^{t} \frac{(1 - \beta_{1,s})\big( \prod_{r=s+1}^{t} \beta_{1,r} \big)}{\sqrt{(1 - \beta_2)\beta_2^{t-s}}} \sqrt{(1 - \beta_2)\beta_2^{t-s}}\, g_{s,i}. \tag{36}$$

Applying Cauchy–Schwarz to Equation (36) and combining Equation (35) yields

$$m_{t,i}^2 \leq \left(\sum_{s=1}^{t} \frac{(1-\beta_{1,s})^2 \left(\prod_{r=s+1}^{t} \beta_{1,r}\right)^2}{(1-\beta_2)\beta_2^{t-s}}\right) \left(\sum_{s=1}^{t}(1-\beta_2)\beta_2^{t-s}g_{s,i}^2\right) = \sum_{s=1}^{t} \frac{(1-\beta_{1,s})^2 \left(\prod_{r=s+1}^{t} \beta_{1,r}\right)^2}{(1-\beta_2)\beta_2^{t-s}} v_{t,i}. \quad (37)$$

Dividing Equation (37) by $\sqrt{v_{t,i}}$ yields $\frac{m_{t,i}^2}{\sqrt{\hat{v}_{t,i}}} \leq \left(\sum_{s=1}^{t} \frac{(1-\beta_{1,s})^2 \left(\prod_{r=s+1}^{t} \beta_{1,r}\right)^2}{(1-\beta_2)\beta_2^{t-s}}\right)\sqrt{v_{t,i}}$. Finally, under Assumption F.2,

it holds that $|g_{s,i}| \leq G_i$ for all $s,i$, and hence Equation (35) implies $v_{t,i} \leq (1-\beta_2)\sum_{s=1}^{t} \beta_2^{t-s}G_i^2 \leq G_i^2$, which yields

$$\begin{aligned}
\sum_{t=1}^{T} A_3 &\leq \sum_{t=1}^{T} \frac{\gamma_t}{2(1-\beta_{1,t})} \left(\sum_{s=1}^{t} \frac{(1-\beta_{1,s})^2 \left(\prod_{r=s+1}^{t} \beta_{1,r}\right)^2}{(1-\beta_2)\beta_2^{t-s}}\right) \sqrt{v_{t,i}} \\
&\leq G_i \sum_{t=1}^{T} \frac{\gamma_t}{2(1-\beta_{1,t})} \left(\sum_{s=1}^{t} \frac{(1-\beta_{1,s})^2 \left(\prod_{r=s+1}^{t} \beta_{1,r}\right)^2}{(1-\beta_2)\beta_2^{t-s}}\right).
\end{aligned} \quad (38)$$

Substituting Equation (30), Equation (33) and Equation (38) into Equation (23), we obtain

$$\begin{aligned}
\sum_{t=1}^{T}\left(\ell_t(w_t) - \ell_t(w^*)\right) &\leq \sum_{i=1}^{d} \frac{D_i^2 G_i}{2\alpha_T(1-\beta_{1,1})} + \left(\sum_{i=1}^{d} D_i G_i\right)\left(\sum_{t=1}^{T} \frac{\beta_{1,t}}{1-\beta_{1,t}}\right) \\
&\quad + \sum_{i=1}^{d}\left[G_i \sum_{t=1}^{T} \frac{\gamma_t}{2(1-\beta_{1,t})}\left(\sum_{s=1}^{t} \frac{(1-\beta_{1,s})^2 \left(\prod_{r=s+1}^{t} \beta_{1,r}\right)^2}{(1-\beta_2)\beta_2^{t-s}}\right)\right].
\end{aligned} \quad (39)$$

Assume that $\beta_{1,t} = \frac{\beta_1}{\sqrt{t}} \in (0,1)$, $\forall t$, and it is non-increasing with the iteration index, i.e., $\beta_{1,1} \geq \beta_{1,2} \geq \cdots \geq \beta_{1,T}$. Therefore, it follows that $\left(\sum_{i=1}^{d} D_i G_i\right)\left(\sum_{t=1}^{T} \frac{\beta_{1,t}}{1-\beta_{1,t}}\right) \leq \left(\sum_{i=1}^{d} D_i G_i\right)\left(\frac{1}{1-\beta_{1,1}}\sum_{t=1}^{T}\beta_{1,t}\right)$, where $\sum_{t=1}^{T}\beta_{1,t} = \beta_1\sum_{t=1}^{T}\frac{1}{\sqrt{t}} \leq \beta_1\left(1 + \int_1^T \frac{1}{\sqrt{x}}\,dx\right) = \beta_1\left(1 + 2(\sqrt{T}-1)\right) \leq 2\beta_1\sqrt{T}$. Then $\prod_{r=s+1}^{t}\beta_{1,r} \leq (\beta_{1,1})^{t-s} = \beta_1^{t-s}$ and $(1-\beta_{1,s})^2 \leq 1$. Hence

$$\left(\sum_{s=1}^{t} \frac{(1-\beta_{1,s})^2 \left(\prod_{r=s+1}^{t} \beta_{1,r}\right)^2}{(1-\beta_2)\beta_2^{t-s}}\right) \leq \sum_{s=1}^{t} \frac{\beta_1^{2(t-s)}}{(1-\beta_2)\beta_2^{t-s}} = \frac{1}{1-\beta_2}\sum_{k=0}^{t-1}\left(\frac{\beta_1^2}{\beta_2}\right)^k.$$

Assume $\beta_1^2 < \beta_2$, the geometric series is bounded by

$$\left(\sum_{s=1}^{t} \frac{(1-\beta_{1,s})^2 \left(\prod_{r=s+1}^{t} \beta_{1,r}\right)^2}{(1-\beta_2)\beta_2^{t-s}}\right) \leq \left(\frac{1}{1-\beta_2}\right)\left(\frac{1}{1-\frac{\beta_1^2}{\beta_2}}\right) = \frac{\beta_2}{(1-\beta_2)(\beta_2-\beta_1^2)}. \quad (40)$$

Substituting Equation (40) into Equation (39), and using the bound $\sum_{i=1}^{d} G_i \leq dG$, we further define the constant $C(\beta_1,\beta_2) \triangleq \frac{\beta_2}{(1-\beta_2)(\beta_2-\beta_1^2)}$, which depends only on the momentum parameters and is finite whenever $\beta_1^2 < \beta_2$. Hence, we obtain

$$\sum_{t=1}^{T}\left(\ell_t(w_t) - \ell_t(w^*)\right) \leq \frac{dD^2 G}{2\alpha_T(1-\beta_1)} + \frac{2dDG\beta_1\sqrt{T}}{1-\beta_1} + \sum_{t=1}^{T}\frac{\gamma_t}{2(1-\beta_1)}dGC(\beta_1,\beta_2).$$

In particular, if $\alpha_T \equiv \gamma_t \equiv \gamma$ is a constant step size, then

$$\sum_{t=1}^{T}\left(\ell_t(w_t) - \ell_t(w^*)\right) \leq \frac{dD^2 G}{2\gamma(1-\beta_1)} + \frac{2dDG\beta_1\sqrt{T}}{1-\beta_1} + T\frac{dG\gamma}{2(1-\beta_1)}C(\beta_1,\beta_2).$$

Dividing both sides by $T$ yields the average optimality gap bound

$$\frac{1}{T}\sum_{t=1}^{T}\big(\ell_t(w_t) - \ell_t(w^*)\big) \leq \frac{dD^2G}{2T\gamma(1-\beta_1)} + \frac{2dDG\beta_1}{(1-\beta_1)\sqrt{T}} + \frac{dG\gamma}{2(1-\beta_1)}C(\beta_1,\beta_2) = O\left(\frac{1}{T\gamma} + \frac{1}{\sqrt{T}} + \gamma\right). \qquad (41)$$

Applying Equation (41) to the $k$-branch and the $v$-branch separately, with possibly different gradient bounds $G_k, G_v$, and using $\ell_t^{\text{total}} = \alpha_k \ell_t^{(k)} + \alpha_v \ell_t^{(v)}$, we obtain

$$\begin{aligned}
\frac{1}{T}\sum_{t=1}^{T}\Big(\ell_t^{\text{total}}(W_{k,t}, W_{v,t}) - \ell_t^{\text{total}}(W_k^*, W_v^*)\Big) \leq \alpha_k &\left[\frac{dD^2G_k}{2T\gamma(1-\beta_1)} + \frac{2dDG_k\beta_1}{(1-\beta_1)\sqrt{T}} + \frac{dG_k\gamma}{2(1-\beta_1)}C(\beta_1,\beta_2)\right] \\
+ \alpha_v &\left[\frac{dD^2G_v}{2T\gamma(1-\beta_1)} + \frac{2dDG_v\beta_1}{(1-\beta_1)\sqrt{T}} + \frac{dG_v\gamma}{2(1-\beta_1)}C(\beta_1,\beta_2)\right].
\end{aligned}$$

