# OpenReview forum: "Semantic-level Backdoor Attack against Text-to-Image Diffusion Models"
_ICML.cc/2026/Conference — ICML 2026 regular_

### Official Review · Reviewer_W9ZT · 2026-02-24

**Soundness:** 4
**Presentation:** 3
**Significance:** 4
**Originality:** 4
**Overall Recommendation:** 5
**Confidence:** 4

**Summary:**

This paper proposes SemBD, a novel attack framework that defines triggers as continuous semantic compositions rather than fixed textual tokens or syntactic patterns. SemBD enables robust and stealthy backdoor activation across semantically equivalent prompts while maintaining model utility and demonstrating strong resistance to existing input-level defenses. SemBD injects semantic-level backdoors by constructing semantic triggers, enforcing activation boundaries through semantic regularization, and aligning cross-attention projections with target representations via a distillation-based editing process. Additionally, experiments on Stable Diffusion v1.5 and SDXL showed that SemBD achieves high attack success rates, strong utility preservation under benign prompts, and improved robustness against input-level defenses such as T2IShield, UFID, and NaviT2I.

**Compliance With Llm Reviewing Policy:**

Affirmed.

**Final Justification:**

The rebuttal has addressed my concerns, and thus I keep my original positive rating.

**Key Questions For Authors:**

See Weaknesses.

**Limitations:**

yes

**Strengths And Weaknesses:**

**Strengths**

* The insights are interesting and reveal the limitations of existing methods, and the proposed SemBD further highlights its advantages by enabling robust and stealthy backdoor activation through semantic-level trigger alignment.
* The paper provides a comprehensive experimental evaluation, comparing SemBD against multiple representative backdoor attack methods (VillanDiffusion, Personalization, Rickrolling, EvilEdit, BadT2I, and IBA) and assessing its robustness under several state-of-the-art defenses (NaviT2I, UFID, and T2IShield). The results clearly demonstrate the effectiveness and robustness of SemBD across diverse settings.
* The paper presents a clear and comprehensive description of the proposed framework, including semantic trigger construction, semantic regularization, multi-entity target design, and projection-level distillation for backdoor injection, with sufficient detail to ensure clarity and reproducib12345ility.
* The paper is well-organized, with a logical progression from motivation to methodology and experimental validation. The figures and explanations effectively illustrate the semantic-level trigger mechanism and cross-attention alignment process, making the overall approach easy to follow.

**Weaknesses**
* The differences between the proposed projection-level distillation mechanism and existing model editing–based backdoor attacks could be clarified more explicitly to better position the contribution.
* The paper could benefit from a brief discussion of practical deployment scenarios, such as potential attack pipelines or real-world applicability, to further emphasize the relevance of the proposed attack.
* The paper demonstrates the effectiveness of semantic-level triggers, but a more detailed discussion on the coverage and sensitivity of the semantic trigger region (e.g., dependence on prompt composition or semantic complexity) could further strengthen the analysis.

---

> ### Author Rebuttal · Authors · 2026-03-31
>
> Thank you very much for your positive evaluation of our paper. We are glad you found the work interesting and clearly written.
>
> Response to Q1.
>
> SemBD is implemented through distillation-based model editing, and its key difference lies in the trigger definition. Prior editing-based attacks are typically tied to discrete trigger forms, whereas SemBD defines the trigger as a continuous compositional semantic region that can be activated by semantically equivalent prompts.
>
> Response to Q2.
>
> A realistic deployment scenario is that an adversary modifies the cross-attention projections of a pre-trained open-source T2I checkpoint and republishes the compromised model. Downstream users may then unknowingly adopt it for image generation, synthetic-data creation, or further fine-tuning, allowing the backdoor to persist and propagate through model reuse.
>
> Response to Q3.
>
> Thank you for this important point. The semantic trigger region in SemBD is local and composition-dependent. Its coverage depends on preserving the full trigger composition, while its sensitivity depends on how much a prompt deviates from that composition. As discussed in our responses to Reviewer DCUE Q2 and Q4, incomplete, semantically adjacent, or unrelated prompts have low or zero FTR. In contrast, semantic triggers reliably activate the backdoor because they preserve the full subject-action-object-scene composition and remain within the local similarity region supported by CLIP proximity and activation behavior.

---

> > ### Author Rebuttal · Reviewer_W9ZT · 2026-04-01
> >
> > My concerns have been addressed.

---

> > > ### Author Response · Authors · 2026-04-01
> > >
> > > Thank you very much for your positive feedback and for confirming that your concerns have been resolved.

---

### Official Review · Reviewer_DCUE · 2026-03-02

**Soundness:** 3
**Presentation:** 3
**Significance:** 3
**Originality:** 3
**Overall Recommendation:** 4
**Confidence:** 4

**Summary:**

This paper introduces Semantic-level Backdoor (SemBD), a novel backdoor attack against text-to-image diffusion models that operates in continuous semantic representation space rather than relying on discrete word- or syntax-level triggers. The key innovation is defining triggers as semantic compositions (subject, action, object, scene) that activate across diverse paraphrases, achieved through distillation-based editing of cross-attention projections while using semantic regularization to prevent unintended activation from incomplete semantics. The authors also propose multi-entity backdoor targets that generate more diverse and realistic poisoned images while further evading detection. The significance of this work lies in demonstrating that semantic-level backdoors achieve 100% attack success rates while reducing state-of-the-art input-level defense detection rates to as low as 2-25.8%, revealing a critical vulnerability gap in current defense mechanisms that focus on discrete input patterns rather than continuous semantic representations.

**Compliance With Llm Reviewing Policy:**

Affirmed.

**Final Justification:**

After considering both the paper and the authors’ rebuttal, I based my final recommendation on an overall assessment of the work’s strengths and weaknesses across soundness, originality, significance, and clarity. In my view, the paper presents a meaningful idea and addresses an interesting problem, with several strengths in its motivation and technical formulation. At the same time, there remain some limitations in aspects such as empirical support, completeness of validation, or the clarity of certain claims and presentation.

The rebuttal was helpful and addressed a number of my concerns, particularly by clarifying the authors’ intent, assumptions, and experimental choices. While it did not completely remove all of my reservations, it improved my understanding of the work and reinforced that the authors had carefully considered the main issues raised during review. Overall, I weighed the paper’s strengths against its remaining weaknesses and provided my final recommendation accordingly. After reading the rebuttal, I increased my score accordingly.

**Key Questions For Authors:**

1. You report that SemBD's FID (23.83) is notably higher (worse) than the benign model's (20.59) and several baseline attacks. Could you provide a more in-depth analysis, such as a qualitative user study or per-prompt diversity metrics, to demonstrate whether this degradation in FID translates to a perceptible loss of image quality or fidelity for end-users on clean prompts? A satisfactory answer showing that the quality drop is imperceptible would strengthen the claim of stealthiness; a confirmation of perceptible degradation would be a significant limitation.

2. To better assess the precision of your semantic regularization, can you provide a quantitative false positive rate—the percentage of prompts containing only partial or incomplete trigger semantics that nonetheless activate the backdoor? Furthermore, an ablation study showing the attack's success rate and false positive rate without this regularization would clarify its exact contribution. This would help determine if the regularization is essential for maintaining specificity or if the projection-level alignment alone already provides sufficient isolation.

3. You claim SemBD is the "first semantic backdoor for T2I diffusion models." Could you clarify how your approach fundamentally differs from prior work that might have implicitly created semantic associations through data poisoning (e.g., poisoning images of "green cars" to always generate a target object), which could also be considered a form of semantic trigger? If such prior work exists, how does your definition of "semantic" (as a continuous, compositional representation edited via model weights) represent a categorical shift rather than an incremental technical improvement?

4. The paper includes a theoretical stability bound (Equation 5) showing that cross-attention output varies smoothly with semantically equivalent prompts. Can you explain how this bound translates to a practical advantage for the attacker? For instance, does it imply a quantifiable "radius" in semantic space within which the backdoor will reliably activate, and how was this radius controlled or measured in your experiments? Connecting this theory more directly to an empirical measurement (e.g., the minimum distance in CLIP space required for activation) would significantly strengthen the paper's contribution.

5. As a weight-poisoning attack that modifies cross-attention projections, have you considered or performed any preliminary experiments to determine whether these modified weights leave a detectable statistical fingerprint? For example, would a simple defense based on pruning or analyzing the norms of the key and value projection matrices be able to flag the model as compromised? A positive answer revealing a vulnerability would be a critical limitation to acknowledge.

**Limitations:**

1. Add a dedicated "Limitations" section that candidly discusses the constraints of SemBD. For example, the method requires white-box access to modify cross-attention parameters, which may not be feasible in all threat models (e.g., when attackers can only poison training data or when models are accessed via APIs).

2. Discuss the computational overhead of generating 11 semantically equivalent prompts and 100 evaluation prompts via GPT-4, as well as the 800 optimization iterations. Acknowledge whether this approach scales efficiently to larger models (e.g., SDXL) or multiple trigger compositions, and whether the attack remains practical under constrained resources.

3. Address the potential for false positives or ambiguous activations when prompts contain semantics that partially overlap with the trigger composition but are not intended to be malicious. For instance, if the trigger involves "cat chasing butterfly," could a prompt about "dog chasing butterfly" inadvertently activate the backdoor? The current t-SNE visualization is insufficient to fully characterize this boundary.

4. Acknowledge that experiments are conducted primarily on standard datasets (e.g., COCO, Pokémon). Discuss whether the attack's effectiveness might degrade on niche domains, low-resource settings, or with significantly different prompt distributions not represented in the evaluation set.

5. Expand the discussion to include potential misuse scenarios (e.g., generating harmful or misleading content at scale) and why the community should care about this vulnerability beyond academic interest. This would help contextualize the significance of the work while maintaining responsible research norms.

**Strengths And Weaknesses:**

Strengths:
+ This paper introduces the conceptually novel and impactful idea of defining backdoor triggers in continuous semantic space rather than at the discrete word or syntax level, shifting the paradigm for both attacks and defenses in text-to-image models.
+ The paper presents significant and compelling results, demonstrating that semantic backdoors achieve near-perfect attack success while rendering state-of-the-art input-level defenses nearly ineffective, with detection rates dropping to as low as 2%.

Weaknesses:
- While the paper reports FID, CLIP score, and LPIPS to demonstrate preserved utility, these metrics are presented in a limited context. The comparison in Table 1 shows that SemBD's FID (23.83) is actually higher (worse) than the benign model's (20.59) and several other attacks, yet this degradation is not discussed. Furthermore, the paper lacks a thorough qualitative analysis or user study to confirm that the generated images from the backdoored model are truly indistinguishable from those of a clean model across a diverse set of benign prompts, which is crucial for establishing stealthiness beyond just evading specific defenses.

- The paper introduces semantic regularization as a critical component to prevent unintended activation from incomplete semantics. However, its effectiveness is only demonstrated through a t-SNE visualization and a single figure (Figure 7) showing reduced activation for substrings. There is no quantitative analysis, such as a false positive rate (i.e., how often a partial-semantic prompt mistakenly triggers the backdoor), nor is there an ablation study showing the consequences of not using this regularization on attack specificity. This makes it difficult to assess how well the method truly isolates the backdoor to the intended full semantic composition.

- The paper positions itself as the "first semantic backdoors for T2I diffusion models." However, prior work in the field of adversarial attacks and model fairness has explored the concept of semantic triggers in other modalities (e.g., vision models) or through data poisoning that alters the semantic understanding of a concept. While the application to T2I diffusion models via model editing is a specific technical contribution, the high-level idea of a "semantic" trigger is less novel than presented. The paper could strengthen its claim by more clearly distinguishing its continuous, compositional, and editing-based approach from any prior work that might have indirectly created semantic associations through poisoned data.

- The paper provides extensive theoretical analysis on the semantic generalization of key/value projections (Appendix B) and the convergence of the distillation optimization (Appendix C). While mathematically sound, these sections feel disconnected from the core empirical findings. The analysis in Appendix B essentially proves that semantically similar prompts lead to similar outputs in a benign model, which motivates the attack but does not provide theoretical guarantees for the attack's success or its ability to evade defenses. The convergence proof in Appendix C is for a simplified, convex version of the optimization, which does not reflect the non-convex nature of the actual loss landscape. This theoretical work adds length without substantially supporting the paper's main claims.

- The paper evaluates SemBD against three input-level defenses (NaviT2I, UFID, T2IShield). However, it does not consider other classes of defenses, such as those based on inspecting the model weights themselves (e.g., pruning, neural cleanse) or those that operate at the output level (e.g., image classifiers for NSFW or specific objects). Since SemBD is a weight-poisoning attack, a comprehensive evaluation should include at least a discussion or basic experiment on whether the modified cross-attention projections leave a detectable statistical fingerprint in the model weights, which could be a significant vulnerability that the paper overlooks.

---

> ### Author Rebuttal · Authors · 2026-03-31
>
> We thank the reviewer for the helpful suggestions.
>
> Response to Q1.
>
> We sincerely apologize for this error. The benign SDv1.5 FID in our paper was misreported, and the correct value is 24.49, which is consistent with prior work (e.g., 24.64 in Clockwork Diffusion [1]). With this correction, the gap to SemBD (23.83) is small. We will release our code to ensure the reproducibility and transparency of this result.
>
> [1] Habibian et al. "Clockwork diffusion: Efficient generation with model-step distillation." CVPR 2024.
>
> Response to Q2.
>
> The effect of different regularization strengths on FTR is discussed in our response to Reviewer cCUi (Q3). Without semantic regularization (λreg = 0), ASR remains 100% on both models, but FTR increases sharply to 77.78% on SDv1.5 and 80.85% on SDXL, showing that semantic regularization is necessary for maintaining specificity.
>
> We quantify false positives using FTR, i.e., the percentage of incomplete-semantic prompts that still activate the backdoor. The results show that FTR is generally low under incomplete semantics.
>
> | Prompt Type              | SDv1.5 FTR (%)  | SDv1.5 CLIP Similarity |  SDXL FTR (%)   | SDXL CLIP Similarity |
> |:------------------------|:----------------:|:----------------:|:--------------:|:--------------:|
> | Semantic Trigger        | –                | 0.81             |  –              | 0.94           |
> | Missing Subject         | 7.0              | 0.66             |  37.8           | 0.91           |
> | Missing Action          | 14.5             | 0.76             |  13.4           | 0.93           |
> | Missing Object          | 3.0              | 0.72             |  0            | 0.89           |
> | Missing Scene           | 17.8             | 0.73             |  30.0           | 0.92           |
> | Two Entities Missing    | 0              | 0.72             |  2.0            | 0.89           |
> | Three Entities Missing  | 0              | 0.40             |  0            | 0.78           |
> | Semantically Adjacent   | 2.0              | 0.64             |  0            | 0.88           |
> | Unrelated Prompt        | 0              | 0.19             |  0            | 0.78           |
>
> Response to Q3.
>
> Prior work may implicitly induce semantic associations, but their trigger condition is still typically tied to discrete textual patterns or data poisoning. In contrast, SemBD explicitly defines the trigger as a compositional semantic region and injects it at the representation level. This is a categorical shift because the trigger is no longer defined in the discrete input space, but as a continuous compositional region in representation space.
>
> Response to Q4.
>
> Thank you for this insightful question. Equation (5) should be interpreted as a local smoothness guarantee rather than a hard threshold. As shown in the table of our response to Q2, CLIP embedding similarity is correlated with FTR. In SDv1.5, incomplete prompts with CLIP similarity above 0.72 tend to have higher FTR, while in SDXL this effect appears above 0.89. These results suggest a measurable local activation region in CLIP space and provide an empirical link between Equation (5) and the minimum semantic distance required for activation.
>
> Response to Q5.
>
> We compared SemBD with benign LoRA fine-tuning and found that simple weight statistics are unreliable because both modify the same cross-attention projections.
> The pruning results on the backdoored SDv1.5 model show that ASR can be reduced, but only at the cost of severe utility degradation. Therefore, simple pruning-based defenses are not a practical way to reliably identify a compromised model.
>
> | Pruning Ratio | 0.1 | 0.2 | 0.3 | 0.4 | 0.5 | 0.6 | 0.7 | 0.8 | 0.9 |
> |--------------|-----|-----|-----|-----|-----|-----|-----|-----|-----|
> | ASR (%)      | 98  | 100 | 97  | 63  | 0   | 0   | 0   | 0   | 0   |
> | LPIPS        | 0.42| 0.43| 0.48| 0.52| 0.58| 0.59| 0.55| 0.54| 0.59|
> | CLIPc         | 24.49 | 24.13 | 23.23 | 21.89 | 17.36 | 18.40 | 20.58 | 20.35 | 16.42 |
> | FID          | 27.06 | 27.16 | 27.26 | 29.52 | 47.53 | 45.10 | 35.46 | 40.33 | 64.48 |
>
>
> Limitations:
>
> 1. The white-box assumption is reasonable in scenarios such as open-source model compromise and third-party fine-tuning services, and most backdoor baselines in our paper follow the same setting, such as EvilEdit (Wang et al., 2024a) and Personalization (Huang et al., 2024). We also acknowledge that this assumption is limiting.
> 2. The attack has some prompt-generation cost, but injection is lightweight and can be run on a single RTX 4090 for both SDv1.5 and SDXL.
> 3. This is already quantified by our FTR-based boundary analysis in Q2, including semantically adjacent prompts.
> 4. We will acknowledge more clearly that SemBD may be less effective on niche domains or under substantially different prompt distributions.
> 5. This is meaningful, and we will make the misuse scenarios more explicit in the revision.

---

> > ### Author Rebuttal · Reviewer_DCUE · 2026-04-03
> >
> > I have no further questions.

---

> > > ### Author Response · Authors · 2026-04-03
> > >
> > > Thanks for your acknowledgment of our work and for your updated assessment. Thanks again for your time and effort in reviewing our paper.

---

### Official Review · Reviewer_cCUi · 2026-03-04

**Soundness:** 3
**Presentation:** 3
**Significance:** 2
**Originality:** 3
**Overall Recommendation:** 5
**Confidence:** 4

**Summary:**

This paper proposes a semantic-level backdoor attack against text-to-image diffusion models, which defines triggers as a semantic composition rather than a fixed word pattern. Specifically, SemBD distills the behavior of cross-attention key and value projections to improve the reliability of backdoor activation by utilizing semantically similar but distinct paraphrases. Experiments on Stable Diffusion models show that SemBD has strong attack effectiveness while maintaining reasonable utility and improved resistance to several input-level detection methods.

**Compliance With Llm Reviewing Policy:**

Affirmed.

**Final Justification:**

The authors propose a novel semantic-level backdoor attack that is well distinguished from traditional backdoor attack paradigms, and validate its effectiveness through well-designed experiments. Therefore, I have increased my score to 5.

**Key Questions For Authors:**

1. How to determine if two prompts are semantically equivalent, and how to prevent semantic drift?

2.  What is the false trigger rate under incomplete semantics?

3. Sweeping the regularization strength and reporting both ASR and false triggering.

4. What are some effective defensive measures?

**Limitations:**

No. The paper should more clearly discuss the limitations and potential negative impact of the attack.

**Strengths And Weaknesses:**

**Strengths**
1. The proposed methods enhance the stealth of the attack, preventing the trigger from being detected by character enumeration.

2. The backdoor embedding process requires minimal modification to the model.

3. The evaluations are comprehensive, covering attack effectiveness, utility maintenance, adaptability under defensive measures, and component ablation.

**Weaknesses**
1. The construction of poisoned trigger prompts is not sufficiently specified. The paper relies on a set of semantically equivalent trigger prompts. However, it does not clearly explain how these prompts are generated, what criteria are used to decide semantic equivalence, or why this particular set is representative.

2. Semantic regularization requires quantitative evaluation (e.g., the rate of unintended trigger; the relationship between regularization strength and this rate), rather than being limited to qualitative analysis.

3. While the authors tested some simple defenses, they did not discuss potential defenses to the proposed attack.

---

> ### Author Rebuttal · Authors · 2026-03-31
>
> Thank you very much for taking the time to evaluate our paper and provide valuable feedback.
>
> Response to Q1.
>
> In SemBD, two prompts are considered semantically equivalent if they jointly satisfy two conditions: (i) preserving the full subject-action-object-scene composition, and (ii) remaining within the local similarity region supported by CLIP proximity and activation behavior. In our paper, Figure 2 shows that prompts with the same underlying semantics are close in both CLIP embedding and projected-value, while Equation (5) shows that prompts close in encoder space remain close in cross-attention output after projection editing, implying a local semantic activation region. The table in our response to Q2 of Reviewer DCUE further supports this view: full semantic triggers have the highest similarity, whereas incomplete or semantically adjacent prompts are less stable and unrelated prompts are the least similar.
>
> To prevent semantic drift, we introduce semantic regularization, which explicitly constrains incomplete semantic variants to remain aligned with the benign model. This regularization is what keeps the trigger boundary local and prevents broad activation under partially matched semantics.
>
> Response to Q2.
>
> Thank you for the question. We have added additional experiments to evaluate the false trigger rate under incomplete semantics. Specifically, we consider 8 types of incomplete semantic prompts. The results show that FTR becomes very low or zero when the semantic composition is further incomplete or unrelated, indicating that SemBD mainly responds to full trigger semantics and that the trigger boundary is local rather than broadly activated by partial semantics.
>
> | Prompt Type            | SDv1.5 FTR (%)   | SDXL FTR (%)   |
> | ---------------------- | ---------------- | -------------- |
> | Missing Subject        | 7.0              | 37.8           |
> | Missing Action         | 14.5             | 13.4           |
> | Missing Object         | 3.0              | 0           |
> | Missing Scene          | 17.8             | 30.0           |
> | Two Entities Missing   | 0              | 2.0            |
> | Three Entities Missing | 0              | 0            |
> | Semantically Adjacent  | 2.0              | 0            |
> | Unrelated Prompts      | 0              | 0            |
>
> Response to Q3.
>
> Thank you for this important question. We sweep λreg and report both ASR and FTR. The results show that semantic regularization is essential. Without it, FTR under incomplete semantics is very high (77.78% on SDv1.5 and 80.85% on SDXL), while adding regularization substantially reduces false triggering and largely preserves ASR. However, overly large λreg hurts the trade-off. Empirically, λreg = 0.5 gives the best balance between attack success and specificity.
>
> | λreg  | SDv1.5 ASR (%)    | SDv1.5 FTR (%)    | SDXL ASR (%)  | SDXL FTR (%)   |
> | ----- | ---------------- | ---------------- | -------------- | -------------- |
> | 0.0   | 100           | 77.78            | 100         | 80.85          |
> | 0.1   | 100           | 27.93            | 100         | 29.79          |
> | 0.2   | 100           | 18.55            | 100         | 17.33          |
> | 0.3   | 100           | 12.16            | 100         | 13.98          |
> | 0.4   | 99.20            | 2.13             | 100        | 10.33          |
> | 0.5   | 100           | 3.04             | 100         | 5.72           |
> | 0.6   | 100           | 7.29             | 93.55          | 1.82           |
> | 0.7   | 94.04            | 1.22             | 94.90          | 3.65           |
> | 0.8   | 97.50            | 3.04             | 95.17          | 6.99           |
> | 0.9   | 98.00            | 1.22             | 72.56          | 10.33          |
> | 1.0   | 95.86            | 0.30             | 84.74          | 10.03          |
>
>
> Response to Q4.
>
> We appreciate this question. An important direction for future research is model-level detection based on anomalies in the model itself, especially detectable statistical signatures and pruning. As discussed in our response to Reviewer DCUE's Q5, pruning can reduce ASR, but only at the cost of severe utility degradation, suggesting that simple pruning-based defenses are not a practical or reliable way to identify a compromised model. We will highlight this as an important future defense direction.

---

> > ### Author Rebuttal · Reviewer_cCUi · 2026-04-01
> >
> > The authors have addressed my concerns, and I have no further comments.

---

> > > ### Author Response · Authors · 2026-04-01
> > >
> > > Thank you again for your time. We are glad that our responses have addressed your concerns.

---

### Official Review · Reviewer_FDhy · 2026-03-12

**Soundness:** 3
**Presentation:** 2
**Significance:** 3
**Originality:** 3
**Overall Recommendation:** 4
**Confidence:** 3

**Summary:**

This paper studies semantic-level backdoor attacks against text-to-image diffusion models. Instead of relying on explicit trigger words or fixed syntactic forms, the proposed SemBD defines the trigger over a semantic composition space and injects the backdoor by aligning cross-attention key/value projections of semantic trigger prompts to those of a target prompt. The method also introduces semantic regularization to suppress partial-trigger activation and a multi-entity target design to reduce attention-pattern consistency. Experiments on Stable Diffusion v1.5 and SDXL show strong attack success and lower detection rates under several input-level defenses.

**Compliance With Llm Reviewing Policy:**

Affirmed.

**Key Questions For Authors:**

1.How stable is SemBD across different semantic trigger/target pairs and random seeds?
2.Why does SemBD preserve utility only moderately on some clean-prompt metrics, and how should this trade-off be interpreted relative to prior baselines?
3.Can the authors provide a trigger-boundary analysis using partially matched, semantically adjacent, and compositional variants of the trigger prompt?

**Limitations:**

No. The paper mentions the attack setting, but the limitations discussion should more clearly acknowledge the restricted evaluation scope, the dependence on a finite semantic trigger set, and the lack of stronger model-side detection baselines.

**Strengths And Weaknesses:**

Strengths. The paper addresses a meaningful and timely threat model beyond word-level and syntax-level backdoors in T2I diffusion models. The method is technically coherent: semantic trigger construction, regularization, multi-entity target design, and projection-level alignment fit together well.The empirical results are strong overall, with 100% ASR and relatively stable stealthiness across multiple input-level defenses.

Weaknesses. 1. The main novelty lies more in the threat formulation and adaptation of existing editing ideas than in a fundamentally new attack mechanism. 2. The claimed “semantic region” is still approximated by a finite set of paraphrased prompts, so the gap between discrete prompt sets and a truly continuous semantic trigger space remains underexplored. 3. The evaluation is still limited in scope: it focuses mainly on a small number of trigger/target constructions and defenses, without enough analysis of trigger boundaries or stability across seeds and semantic pairs.

---

> ### Author Rebuttal · Authors · 2026-03-31
>
> We thank the reviewer for all the insightful comments. We have addressed your questions and comments below.
>
> Response to Q1.
>
> We evaluate SemBD using 5 different trigger–target pairs and 6 random seeds. The results show that SemBD is stable across different trigger-target pairs and random seeds on both SDv1.5 and SDXL.
>
> | Pair | Trigger | Target |
> |------|---------|--------|
> | 1 | A bird is sitting on a branch in the tree | A firefighter wearing red clothing is spraying water onto a house |
> | 2 | A man on the street is holding an umbrella | A pizza is sitting on a table along with a drink |
> | 3 | A dog is chasing a ball in the park | A soldier holding a rifle wearing a helmet in the battlefield |
> | 4 | The cat in the yard chased a butterfly | An astronaut riding a horse wearing a backpack on Mars |
> | 5 | A bird is sitting on a branch in the tree | A child holding a gun wearing a hat in the school |
>
> | Pair | ASR (SDv1.5) |CLIPp (SDv1.5)  | ASR (SDXL)  |CLIPp (SDXL)|
> | -------------- | --------- | --------- | --------- | --------- |
> |  1         | 100       | 29.46     | 98.96     | 27.86     |
> |  2         | 100       | 28.86     | 99.20     | 28.77     |
> |  3         | 100       | 25.36     | 100       | 25.58     |
> |  4         | 99.7      | 26.90     | 100       | 27.30     |
> |  5         | 100       | 27.61     | 100       | 27.68     |
> | mean           | 99.94     | 27.64     | 99.63     | 27.44     |
>
>
> |Seed (SDv1.5)| ASR  (%)  | CLIPp    | LPIPS   | CLIPc    | FID      | NaviT2I   | UFID      | T2IShieldFTT | T2IShieldCDA |
> | -------- | --------- | --------- | -------- | --------- | --------- | --------- | --------- | ------------ | ------------ |
> | 42       | 100     | 28.16     | 0.34     | 25.68     | 23.87     | 12.60     | 17.65     | 30.80        | 2.00         |
> | 67       | 99.8      | 28.20     | 0.30     | 25.66     | 23.82     | 14.20     | 20.18     | 25.40        | 2.00         |
> | 456      | 100     | 28.30     | 0.31     | 25.62     | 23.44     | 11.95     | 21.60     | 30.55        | 0         |
> | 678      | 100   | 28.09     | 0.33     | 25.71     | 23.83     | 12.00     | 15.00     | 36.40        | 10.20        |
> | 1000     | 100     | 27.73     | 0.31     | 25.73     | 23.61     | 11.00     | 18.90     | 25.60        | 6.45         |
> | 11726    | 99.9      | 28.19     | 0.34     | 25.60     | 23.71     | 19.50     | 16.85     | 12.00        | 3.00         |
> | mean     | 99.95     | 28.11     | 0.32     | 25.67     | 23.71     | 13.54     | 18.36     | 26.79        | 3.94         |
>
> | Seed (SDXL)  | ASR  (%)     | CLIPp    | LPIPS    | CLIPc     | FID       |
> | ------------ | --------- | --------- | -------- | --------- | --------- |
> | Benign Model | –         | 7.13      | 0.00     | 26.21     | 29.79     |
> | 42           | 100       | 28.40     | 0.27     | 25.63     | 30.19     |
> | 67           | 100       | 28.08     | 0.28     | 25.77     | 30.56     |
> | 456          | 100       | 28.20     | 0.27     | 25.71     | 30.24     |
> | 678          | 99.6      | 28.19     | 0.29     | 25.77     | 30.39     |
> | 1000         | 99.3      | 27.93     | 0.26     | 25.78     | 30.34     |
> | 11726        | 100       | 28.15     | 0.26     | 25.67     | 30.37     |
> | mean         | 99.82     | 28.16     | 0.27     | 25.72     | 30.35     |
>
> Response to Q2.
>
> Thank you for raising this concern. We sincerely apologize for this error. The benign SDv1.5 FID was misreported in our paper, and the correct value is 24.49, which is consistent with prior work (e.g., 24.64 reported in Clockwork Diffusion [1]). We will release our code to ensure the reproducibility and transparency of this result.
>
> [1] Habibian et al. "Clockwork diffusion: Efficient generation with model-step distillation." CVPR 2024.
>
> Response to Q3.
>
> Thank you for raising this. We analyze the trigger boundary using False Trigger Rate (FTR), i.e., the probability that incomplete semantic prompts activate the backdoor. The results show that the trigger boundary is local and composition-dependent.
>
> | Prompt Type            | SDv1.5 FTR (%)  | SDXL FTR (%)  |
> | ---------------------- | ---------------- | -------------- |
> | Missing Subject        | 7.0              | 37.8           |
> | Missing Action         | 14.5             | 13.4           |
> | Missing Object         | 3.0              | 0            |
> | Missing Scene          | 17.8             | 30.0           |
> | Two Entities Missing   | 0              | 2.0            |
> | Three Entities Missing | 0              | 0           |
> | Semantically Adjacent  | 2.0              | 0            |
> | Unrelated Prompts      | 0              | 0            |

---

> > ### Author Rebuttal · Reviewer_FDhy · 2026-04-03
> >
> > I have no further questions.

---

> > > ### Author Response · Authors · 2026-04-03
> > >
> > > We sincerely appreciate your positive feedback and are glad that your concerns have been addressed.

---

### Decision · Program_Chairs · 2026-04-30

**Decision:**

Accept (regular)

**Comment:**

This paper proposes SemBD, a semantic-level backdoor attack against T2I diffusion models that defines triggers as continuous compositional semantic regions in representation space, injects backdoors via distillation-based editing of cross-attention key/value projections, and employs semantic regularization and multi-entity targets to enhance stealthiness. Experiments on SD v1.5 and SDXL demonstrate very high ASR with strong robustness against state-of-the-art defenses including T2IShield, UFID, and NaviT2I. The paper received reviews from four reviewers (W9ZT, FDhy, cCUi, DCUE). Reviewer W9ZT praised the comprehensive evaluation and clear presentation but requested clarification on distinctions from prior editing-based attacks and deeper trigger coverage analysis. Reviewer FDhy acknowledged the meaningful threat model but raised concerns about limited novelty, the gap between discrete paraphrase sets and a truly continuous trigger space, and insufficient stability analysis. Reviewer cCUi (Accept) highlighted enhanced stealthiness and minimal model modification, but noted insufficient specification of poisoned prompt construction and lack of quantitative regularization evaluation. Reviewer DCUE recognized the novel semantic-level trigger paradigm but raised questions about semantic equivalence determination and false trigger rates. During rebuttal, the authors addressed all major concerns: providing stability results, introducing False Trigger Rate (FTR) analysis to empirically ground the local activation region, quantitatively evaluating the regularization strength trade-off, and discussing pruning-based defenses and deployment scenarios. All four reviewers confirmed their concerns were fully resolved. I recommend accepting this paper.